# A cluster-assisted differential evolution-based hybrid oversampling method for imbalanced datasets

Muhammed Abdulhamid Karabiyik[1], Bahaeddin Turkoglu[2] and Tunc Asuroglu[3,4]

[1] Department of Computer Engineering, Nigde Omer Halisdemir University, Nigde, Turkey
[2] Department of Artificial Intelligence and Data Engineering, Ankara University, Ankara, Turkey
[3] Faculty of Medicine and Health Technology, Tampere University, Tampere, Finland
[4] VTT Technical Research Centre of Finland, Tampere, Finland

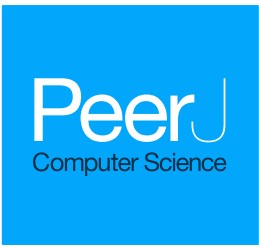

Corresponding author
Tunc Asuroglu,
tunc.asuroglu@tuni.fi

## ABSTRACT

Class imbalance remains a significant challenge in machine learning, leading to biased models that favor the majority class while failing to accurately classify minority instances. Traditional oversampling methods, such as Synthetic Minority Over-sampling Technique (SMOTE) and its variants, often struggle with class overlap, poor decision boundary representation, and noise accumulation. To address these limitations, this study introduces ClusterDEBO, a novel hybrid oversampling method that integrates K-Means clustering with differential evolution (DE) to generate synthetic samples in a more structured and adaptive manner. The proposed method first partitions the minority class into clusters using the silhouette score to determine the optimal number of clusters. Within each cluster, DE-based mutation and crossover operations are applied to generate diverse and well-distributed synthetic samples while preserving the underlying data distribution. Additionally, a selective sampling and noise reduction mechanism is employed to filter out low-impact synthetic samples based on their contribution to classification performance. The effectiveness of ClusterDEBO is evaluated on 44 benchmark datasets using k-Nearest Neighbors (kNN), decision tree (DT), and support vector machines (SVM) as classifiers. The results demonstrate that ClusterDEBO consistently outperforms existing oversampling techniques, leading to improved class separability and enhanced classifier robustness. Moreover, statistical validation using the Friedman test confirms the significance of the improvements, ensuring that the observed gains are not due to random variations. The findings highlight the potential of cluster-assisted differential evolution as a powerful strategy for handling imbalanced datasets.

## INTRODUCTION

Class imbalance is a pervasive issue in machine learning, significantly affecting the performance of classification models by biasing predictions toward the majority class while neglecting the minority class (*Haixiang et al., 2017*). This challenge is particularly critical

in high-impact applications such as medical diagnosis, fraud detection, cybersecurity, and industrial fault detection, where accurate recognition of rare instances is essential (*Karaaslan et al., 2024*; *Sun et al., 2020*; *Zareapoor & Yang, 2017*). Traditional approaches to handling imbalanced datasets can be broadly categorized into algorithmic-level modifications and data-level resampling techniques (*Liu, Fan & Wu, 2019*). While algorithmic solutions attempt to adjust decision thresholds or incorporate cost-sensitive learning strategies, data-level solutions—particularly oversampling methods—remain the most widely used due to their direct impact on training data representation (*López et al., 2013*).

Among oversampling methods, the Synthetic Minority Oversampling Technique (SMOTE) and its variants have been extensively utilized to generate synthetic samples for the minority class. These methods create interpolated samples between existing minority instances, increasing their representation without discarding majority class data. However, SMOTE-based methods often suffer from significant limitations, such as generating synthetic samples in sparse or overlapping regions, leading to class overlap and noise accumulation. Moreover, traditional oversampling techniques typically ignore the distributional characteristics of minority class instances, treating all minority samples equally without prioritizing the most informative ones (*Chawla et al., 2008*; *Tao, Wang & Zhang, 2019*).

These methods are particularly prone to generating synthetic samples in sparsely populated or overlapping regions, which often leads to decision boundary distortion and noise accumulation. Such behavior can severely degrade classification performance, especially when minority instances exhibit non-linear or complex boundary structures. This motivates the need for a more adaptive and structure-aware oversampling strategy that can effectively handle sparse distributions while preserving class separability (*Uymaz et al., 2024*).

In parallel with clustering-assisted oversampling approaches, recent research has explored the use of Dempster–Shafer theory to address uncertainty and class ambiguity in imbalanced learning. These methods assign belief masses to instances or clusters and utilize evidence fusion strategies to guide the oversampling process more reliably, particularly in unsupervised or noise-prone settings (*Lin & Leony, 2024*; *Tian et al., 2024*). While conceptually distinct from optimization-based techniques, Dempster–Shafer based models have proven valuable in modeling soft cluster boundaries and handling class overlap, thereby enriching the landscape of data level solutions for imbalanced classification.

To address these limitations, cluster-based oversampling methods have been proposed, where minority class samples are first grouped into clusters before synthetic samples are generated within each cluster. This structured approach helps preserve local data distributions, ensuring that newly generated samples better reflect the actual minority class structure. Additionally, evolutionary optimization techniques, particularly differential evolution (DE) (*Storn & Price, 1997*), have demonstrated remarkable potential in improving oversampling effectiveness by introducing controlled perturbations and generating diverse yet meaningful synthetic instances. Despite these advancements, a

fundamental challenge remains existing oversampling methods lack an integrated framework that combines clustering-based sample distribution analysis with an adaptive, evolution-driven oversampling mechanism.

In this study, we introduce ClusterDEBO, a novel hybrid oversampling method that integrates K-Means (*Ahmed, Seraj & Islam, 2020*) clustering with DE-based synthetic sample generation to enhance the robustness of imbalanced learning. ClusterDEBO leverages K-Means clustering to segment the minority class into subgroups, ensuring that synthetic samples are generated in a manner that preserves the natural distribution of data. This clustering step prevents excessive sample generation in sparse regions and ensures that the new samples contribute effectively to classifier performance. After clustering, a DE-based perturbation strategy is applied to generate synthetic samples that maintain intra-cluster diversity while avoiding redundant or misleading samples. The generated synthetic samples are then subjected to noise reduction and selective sampling mechanisms, where only those instances that positively impact classification performance, measured using area under curve (AUC) based impact assessment, are retained.

To evaluate the effectiveness of ClusterDEBO, extensive experiments are conducted on 44 benchmark datasets (*Alcalá-Fdez et al., 2009*), comparing it against widely used oversampling methods, including Borderline-SMOTE1 (*Han, Wang & Mao, 2005*), Borderline-SMOTE2 (*Han, Wang & Mao, 2005*), Safe-Level-SMOTE (*Bunkhumpornpat, Sinapiromsaran & Lursinsap, 2009*), SMOTE-edited nearest neighbors (S-ENN) (*Wilson, 1972*), Adaptive-SMOTE (ADASYN) (*He et al., 2008*), S-RSB (*Ramentol et al., 2012*), SMOTE-Tomek Links (*Tomek, 1976*), and differential evolution algorithm for highly imbalanced datasets (DEBOHID) (*Kaya et al., 2021*). The classification performance is assessed using support vector machine (SVM), decision tree (DT), and k-nearest neighbor (kNN), with area under the curve (AUC) (*Myerson, Green & Warusawitharana, 2001*) serving as the primary evaluation metrics. Furthermore, Friedman statistical testing is performed to validate the robustness of the results, ensuring that improvements are consistent and not due to random variations (*Gibbons, 1993*).

By incorporating distribution-aware clustering assisted evolutionary-inspired synthetic sample generation, ClusterDEBO addresses key challenges in imbalanced learning, providing a more adaptive, noise-resilient, and structurally coherent oversampling framework. The results demonstrate that ClusterDEBO significantly enhances classifier robustness, reduces class overlap, and improves minority class recognition, offering a powerful and computationally efficient solution for handling highly imbalanced datasets.

The remainder of this article is organized as follows: 'Materials and Methods' provides a detailed explanation of the materials and methods, including the proposed ClusterDEBO approach, the integration of K-Means clustering, and the DE-based synthetic data generation process. 'Experimental Setup' presents the experimental setup, describing the datasets, evaluation metrics, and baseline comparisons used to assess the effectiveness of ClusterDEBO. 'Results' reports the results obtained from comparative evaluations, while 'Discussion' discusses the implications of these results, including performance insights and methodological advantages. Finally, 'Conclusions and Future Works' concludes the study, summarizing key contributions and outlining potential directions for future research.

## MATERIALS AND METHODS

Imbalanced datasets represent a critical challenge in machine learning that significantly undermines the performance of classification algorithms. While numerous oversampling techniques have been developed to enhance the representational capacity of minority classes, these methods typically rely on simplistic strategies such as random interpolation or basic replication approaches.

This research introduces ClusterDEBO, a novel methodology that integrates K-Means clustering with DE to generate a more sophisticated and decision boundary-aware oversampling process. The proposed approach distinguishes itself through a comprehensive framework comprising cluster-based sampling, hybrid synthetic data generation using DE, noise reduction techniques, and selective sampling mechanisms.

The methodology addresses the fundamental limitations of existing oversampling strategies by leveraging advanced computational intelligence techniques. By dynamically adapting to the underlying data distribution, ClusterDEBO aims to mitigate the representational disparities inherent in imbalanced datasets, thereby improving the generalizability and predictive performance of machine learning models.

### K-means clustering algorithm

The K-Means algorithm is a popular clustering method that divides data into a certain number of clusters based on their similarities (*Ahmed, Seraj & Islam, 2020*). It is particularly well-suited for large-scale datasets due to its efficient use of computing resources. Within the scope of the ClusterDEBO method, the K-Means algorithm is utilized to categories minority class examples into subgroups. This approach ensures that each cluster exhibits a more homogeneous distribution, thereby enhancing the sensitivity of synthetic data generation to intra-cluster features.

The K-Means algorithm commences with the selection of an initial number of centers ($k$), specified. Subsequently, each data point is allocated to a cluster based on the proximity to the nearest center point. Subsequent to this assignment process, the new center point for each cluster is updated by taking the average of the data belonging to the cluster. The centroid of each cluster is computed using the mean of all data points assigned to that cluster, as defined in Eq. (1).

$$c_j = \frac{1}{|S_j|} \sum_{x \in S_j} x. \tag{1}$$

Here, $S_j$ represents the set of data points in the $j$-th cluster. Once the new centroids are computed, the data points are reassigned to their nearest centroid, updating the clusters accordingly. The algorithm terminates when the centroid positions remain unchanged or when the predefined iteration limit is reached.

The success of the K-Means algorithm largely depends on determining the appropriate number of clusters. In the ClusterDEBO framework, the silhouette score method is used to determine the optimal number of clusters. The silhouette score helps assess the quality of clustering by measuring both intra-cluster consistency and inter-cluster separation. It is

calculated using a normalized formula that considers the average distance of a data point to other points within the same cluster (a) and the average distance to the nearest neighboring cluster (b). The clustering quality is then assessed using the silhouette score, calculated *via* Eq. (2), which captures intra-cluster cohesion and inter-cluster separation.

$$S = \frac{b - a}{\max(a, b)}.$$

(2)

Here, the S score ranges between −1 and 1. A S = 1 value indicates that the data point is well clustered, S = 0 suggests that the point lies on the boundary between clusters, and S < 0 signifies that the point has been incorrectly clustered. A high silhouette score indicates a well-structured clustering where data points are correctly assigned to their respective clusters. Accurately determining the number of clusters ensures a more balanced synthetic data generation process. The ClusterDEBO method utilizes K-Means clustering to segment the minority class into subgroups, thereby increasing the sensitivity of generated synthetic data to decision boundaries.

## Differential evolution algorithm

DE is a population-based metaheuristic optimization algorithm designed to solve continuous optimization problems. It operates through four main stages: initialization, mutation, crossover, and selection. Each candidate solution (also referred to as a vector) in the population is represented in a D-dimensional search space.

Let the population consist of N individuals, where each individual is a vector $X_i = [X_{i,1}, X_{i,2}, \ldots, X_{i,D}]$, *for* $i = 1, 2, \ldots, N$. The initialization step generates these individuals randomly within specified bounds using:

$$X_{i,j} = Min_j + rand(0, 1) \times (Max_j - Min_j) \quad \forall j \in \{1, 2, \ldots, D\}.$$

(3)

Here:

$X_{i,j}$ : the value of the $j^{\text{th}}$ dimension of the $i^{\text{th}}$ individual,

$Min_j$ and $Max_j$: the lower and upper bounds for dimension $j$,

$rand(0, 1)$: a uniformly distributed random number between 0 and 1.

Once initialized, DE proceeds with the mutation step, where a donor vector $V_i$ is generated by combining three distinct individuals $X_{r1}, X_{r2}, X_{r3}$, randomly selected from the current population ($r1 \neq r2 \neq r3 \neq i$). Mutation is performed according to the standard DE/rand/1 scheme, as shown in Eq. (4).

$$V_i = X_{r1} + F \cdot (X_{r2} - X_{r3}).$$

(4)

Here, $F \in [0, 2]$ is the scaling factor that controls the amplification of the differential variation.

The crossover step creates a trial vector $T_i = [T_{i,1}, T_{i,2}, \ldots, T_{i,D}]$ by mixing the donor vector $V\_i$ and the original individual $X\_i$ according to the crossover rate $CR \in [0, 1]$. The crossover mechanism is defined formally in Eq. (5), ensuring at least one component is inherited from the donor.

$$T_{i,j} = \begin{cases} V_{i,j}, & if\ rand(0,1) \leq CR\ or\ j = j_{\text{rand}} \\ X_{i,j}, & otherwise \end{cases} \tag{5}$$

where:

$j_{\text{rand}} \in \{1, \ldots, D\}$ is a randomly chosen index to ensure that at least one component comes from $V_i$,

$rand(0,1)$: a uniformly distributed random number.

Finally, the selection step compares the trial vector $T_i$ with the original individual $X_i$. If $T_i$ yields a better value for the objective function, it replaces $X_i$ in the next generation. Otherwise, $X_i$ is retained.

In this study, DE is adapted for synthetic data generation inspired by the DE/rand/1 scheme, where new minority class instances are created by evolving existing minority samples using the operations described above. This controlled yet stochastic generation mechanism ensures both diversity and fidelity to the underlying data structure.

In the context of synthetic sample generation *via* DE, the choice of DE strategy and its associated control parameters, mutation factor (F) and crossover rate (CR) plays a crucial role in determining the quality and diversity of the generated instances. To ensure that our proposed method builds on a robust and empirically validated foundation, we adopted the DE variant identified as DSt1, which corresponds to the DE/rand/1 scheme. This decision was grounded in the findings of *Korkmaz et al. (2021)*, who conducted a comprehensive evaluation of 16 DE-based oversampling strategies on 44 imbalanced datasets using three different classifiers (SVM, kNN, DT). Their study demonstrated that DSt1 achieved the best or near-best AUC and G-Mean performance across most datasets, indicating its superior ability to balance exploration and exploitation in the context of imbalanced learning. By incorporating this well-performing DE variant into ClusterDEBO, we aim to leverage an already optimized parameter configuration and direct our methodological focus toward enhancing sample distribution and boundary fidelity through clustering and selective sampling mechanisms. This ensures that our contribution remains focused and does not duplicate well-established findings on DE strategy comparisons.

Furthermore, although dataset-specific hyperparameter tuning could potentially yield marginal performance gains, such a step was not repeated in this study to avoid confounding the effects of clustering integration and DE-based oversampling. Nonetheless, we acknowledge this direction as a valuable avenue for future research and discuss it in the conclusion section.

## Proposed method: clusterDEBO

The ClusterDEBO method is a structured oversampling approach designed to enhance the representation of minority class instances in imbalanced datasets. The methodology follows a systematic data processing and model development pipeline, consisting of dataset preparation, partitioning into training and test sets, synthetic data generation using the ClusterDEBO approach, model training, and performance evaluation. The overall workflow of these stages is illustrated in Fig. 1.

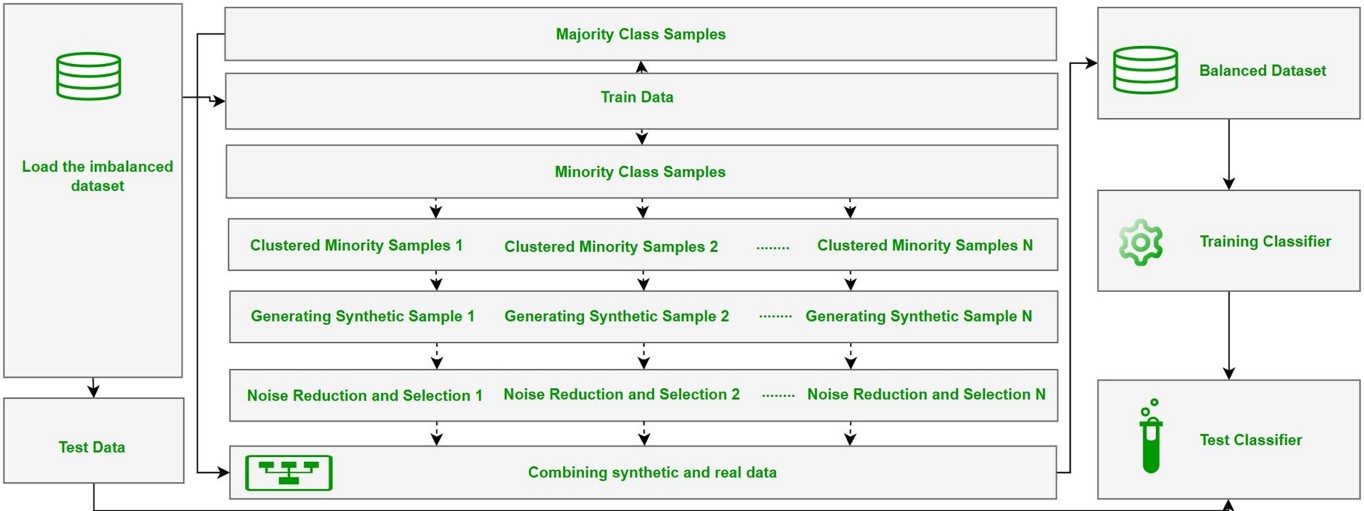

**Figure 1** The overall workflow of the stages.

The ClusterDEBO method combines the K-Means clustering algorithm with the DE-based DEBOHID method, allowing synthetic data to be generated in a more balanced and structurally appropriate manner in the data space.

This process includes the steps of clustering minority class samples, determining appropriate data points in each cluster, creating synthetic samples with the DE mechanism, and selective sampling with noise reduction in the final stage. The pseudo code of the ClusterDEBO method is Algorithm 1.

### Determination of minority and majority classes

The first step of the proposed method is to identify the minority and majority classes in the dataset. Minority class instances are usually limited in number and dominated by the majority class. Therefore, it is important to understand the structure of the dataset and identify the imbalance. Figure 2 shows examples of minority (red) and majority (grey) classes in the dataset.

### Separating minority class into subsets with k-means algorithm

In the second step of ClusterDEBO, minority class samples are divided into a certain number of clusters using the K-Means clustering method. The number of clusters is determined by optimizing with the silhouette score method (*Shahapure & Nicholas, 2020*). Clustered minority class samples allow the determination of the centroids for each cluster, and these centroids are used as reference points in synthetic data generation. However, to prevent instability in clustering when the minority class has very few instances, a threshold mechanism is applied. If the silhouette score falls below 0.25, clustering is bypassed by setting the number of clusters to 1, thereby avoiding artificial or noisy cluster formations (*Lovmar et al., 2005*). This fallback approach ensures robustness and effectively reverts the system to the behavior of the original DEBOHID method in such low-sample scenarios. Figure 3 illustrates the distribution of the dataset after the subsetting process.

| Algorithm 1 ClusterDEBO. |
| --- |

```
1   Input: Imbalanced dataset (X, y), Number of clusters (k),
2   Differential Evolution parameters (F, CR, Number of generations)
3   Output: Balanced dataset (X_new, y_new)
4
5   # Step 1: Identify minority class samples
6   X_min = []
7   for each sample x in X:
8       if y[x] == minority_class:
9           X_min.append(x)
10
11  # Step 2: Perform K-Means clustering on minority class samples
12  clusters = k_means(X_min, k)
13
14  # Step 3: Generate synthetic samples using Differential Evolution
15  for each cluster in clusters:
16      center = compute_cluster_center(cluster)
17
18      for i in range(len(cluster)):
19          # Select three random samples from the cluster
20          x_r1, x_r2, x_r3 = select_random_samples(cluster, 3)
21
22          # Apply Differential Evolution mutation
23          v_i = x_r1 + F * (x_r2 − x_r3)
24
25          # Perform crossover operation
26          u_i = crossover (v_i, x_r1, CR)
27
28          # Validate and add the synthetic sample
29          if is_valid_sample(u_i):
30              X_new.append(u_i)
31
32  # Step 4: Outlier detection and removal based on cluster centers
33  for each sample x in X_new:
34      if is_outlier(x, clusters):
35          X_new.remove(x)
36
37  # Step 5: Noise reduction and selective sampling
38  for each sample x in X_new:
39      if evaluate_auc_impact(x, model) < threshold:
40          X_new.remove(x)
41  return X_new, y_new
```

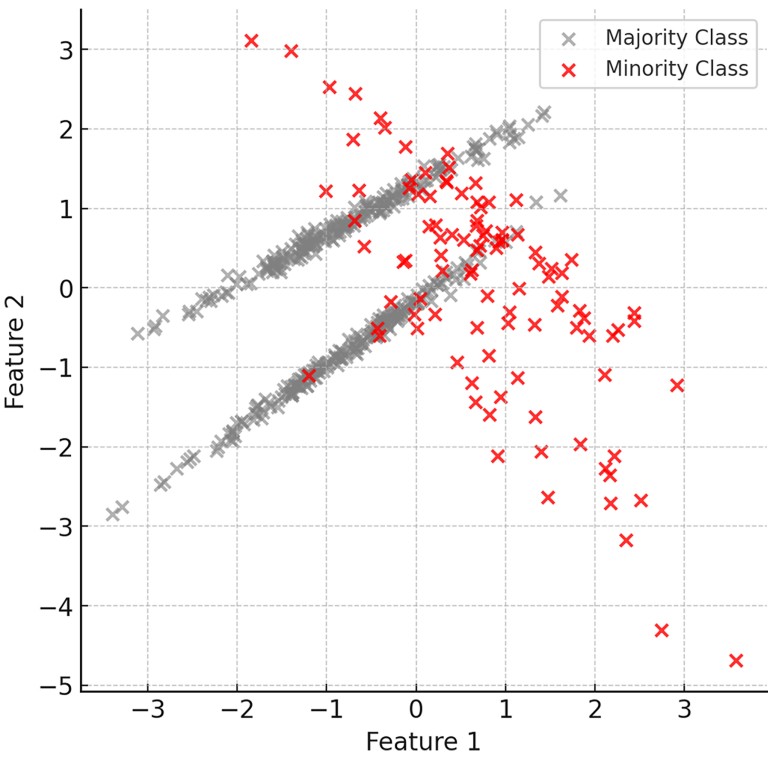

**Figure 2 Examples of minority (red) and majority (grey) classes in the dataset.**

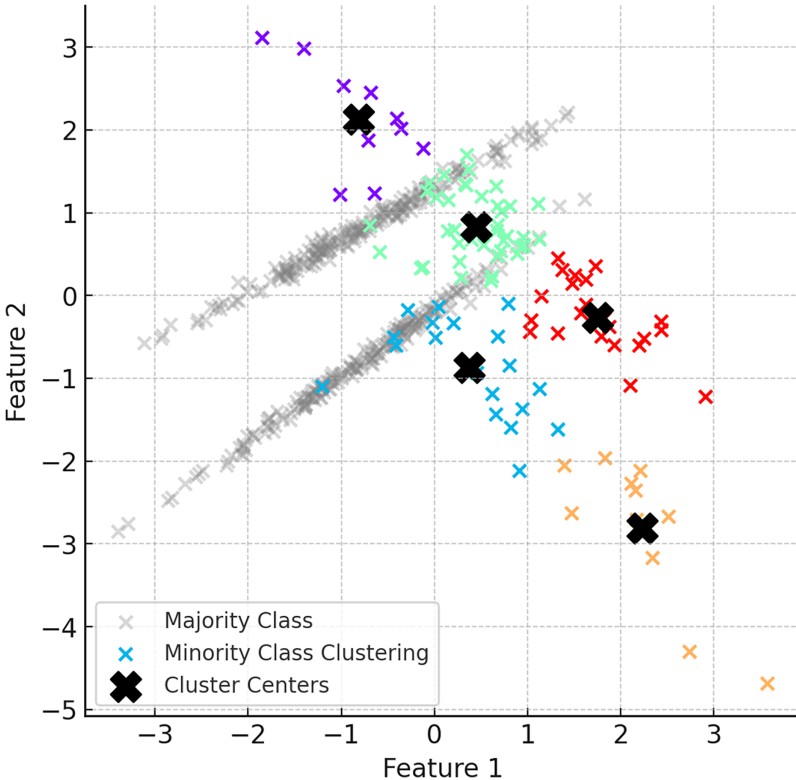

**Figure 3 Visual of the dataset after subsetting process.**

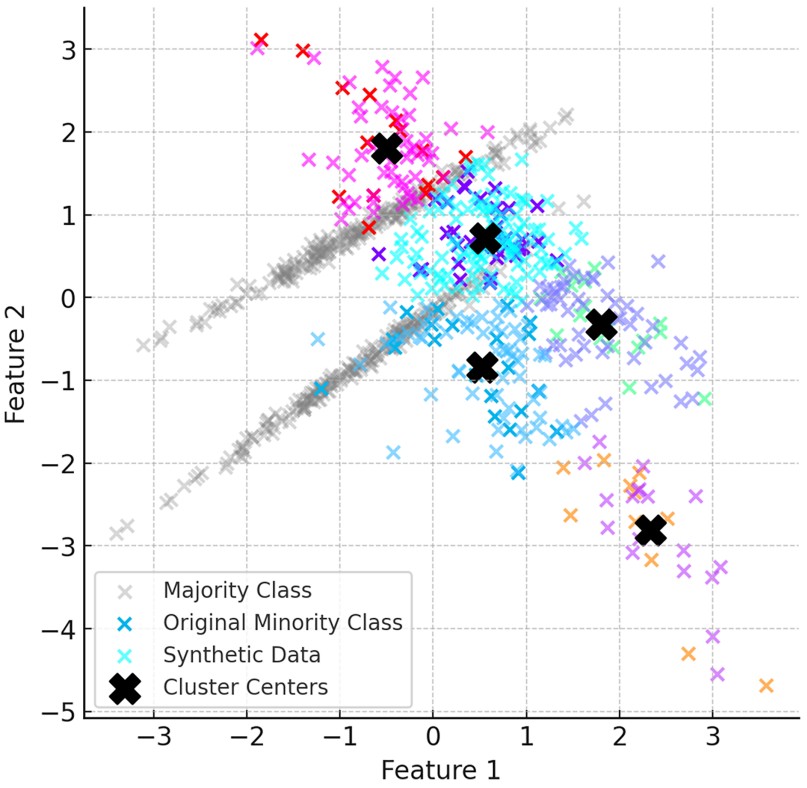

**Figure 4 Data generated with DEBOHID.**

### Synthetic data generation with DE

After the K-Means clustering process is completed, synthetic data are generated for each cluster using the DE based DEBOHID method. The generated synthetic data are added in a balanced way to show a more homogeneous distribution around the cluster centres. With the addition of synthetic data, the clustering process is repeated and the cluster centres are updated. Figure 4 shows the data generated with DEBOHID.

### Noise reduction and selective sampling

The generation of synthetic data in ClusterDEBO is bounded by the spatial limits of each cluster to ensure local coherence and avoid the creation of unrealistic or outlier instances. Specifically, each synthetic sample is evaluated based on its distance from the corresponding cluster center and is accepted only if it falls within the maximum radius observed among the real minority samples in that cluster. If a valid sample cannot be generated within this boundary after a predefined number of attempts (*e.g.*, 10 trials), the last generated sample is rescaled toward the cluster center so that it fits within the radius. This mechanism not only ensures the structural integrity of the generated data but also prevents the risk of infinite loops during sample generation. Thus, the method integrates a boundary-aware control mechanism that preserves class locality while maintaining generation stability. Figure 5 provides a visual summary of this controlled sampling and noise limiting process.

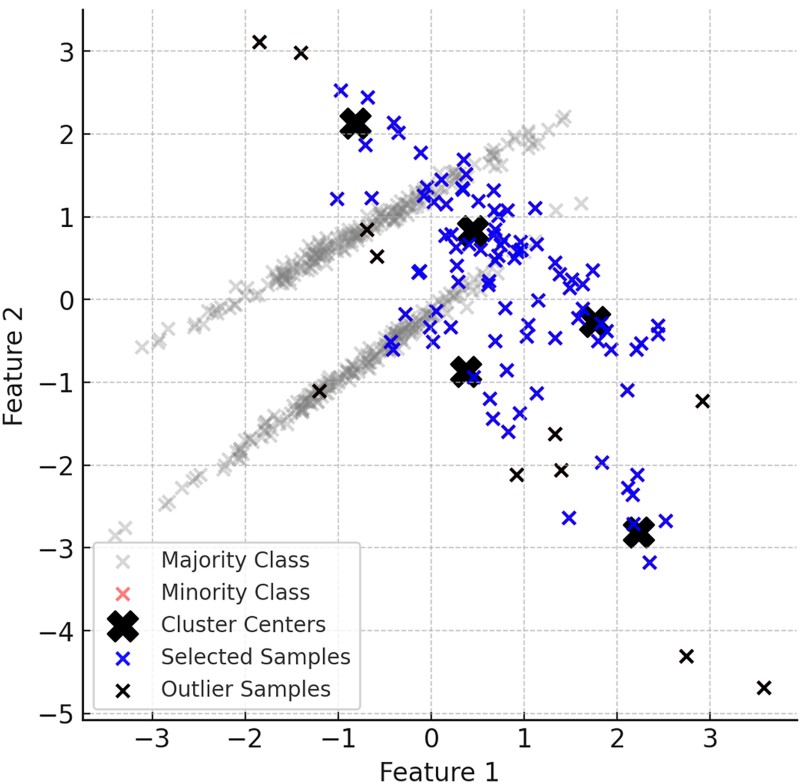

**Figure 5** A visual representation of the noise reduction and data selection process.

### Final balanced dataset

When the ClusterDEBO method is complete, the minority class instances have been increased in a balanced way and are more aligned with the majority class. Figure 6 shows the final balanced dataset and the synthetic data produced.

Compared to the classical DEBOHID method, the ClusterDEBO method aims to increase the generalisation success of the model by making the synthetic data generation process more controlled by clustering. The clustering process prevents the occurrence of extreme outliers by better adapting to the natural distribution of minority class samples. The DE algorithm ensures that the synthetic data produced is more realistic by taking into account intra-cluster variations. Noise reduction and selective sampling steps guarantee the selection of the most appropriate data by evaluating the contribution of synthetic samples added to the dataset to the model performance. Thus, the ClusterDEBO method offers an oversampling strategy that both better represents decision boundaries and improves classification performance.

## Experimental setup

To assess the effectiveness of the proposed ClusterDEBO method, extensive experiments were conducted on 44 publicly available benchmark datasets from the KEEL repository (*Alcalá-Fdez et al., 2009*). These datasets are widely recognized as standard benchmarks in

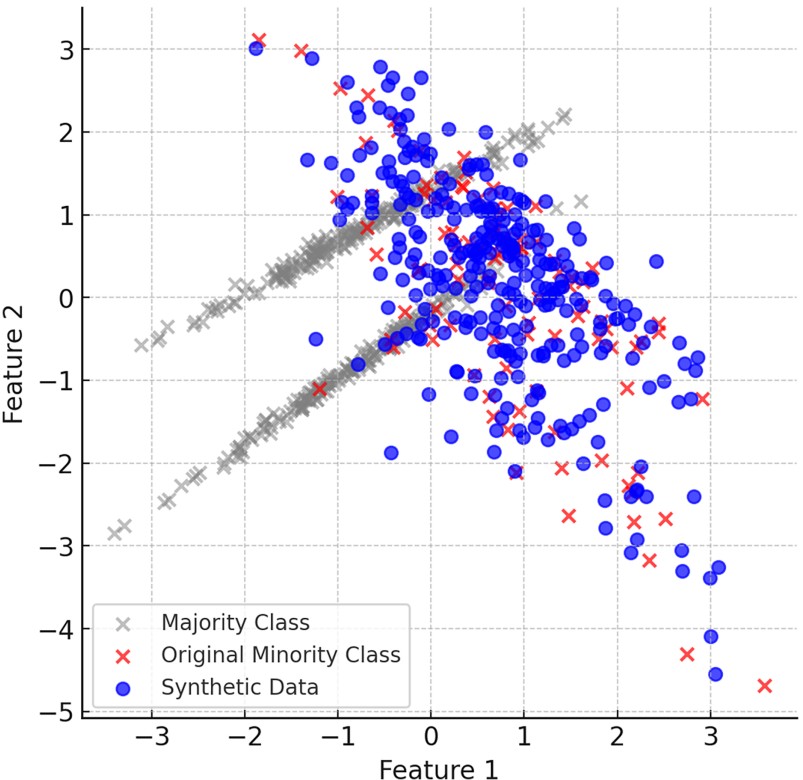

**Figure 6 The final balanced dataset and the synthetic data produced.**

the literature for evaluating imbalanced classification techniques (*Alcalá-Fdez et al., 2015*; *Triguero et al., 2017*). They encompass a diverse range of real-world applications, including medical diagnostics, fraud detection, cybersecurity, fault prediction, and text classification. The inclusion of datasets from multiple domains ensures that the proposed method is tested across various levels of class imbalance and feature complexity, providing a rigorous and comprehensive evaluation framework.

Table 1 presents an overview of the datasets used in this study, detailing key characteristics such as the total number of instances, feature dimensions, the proportion of minority class samples, and the imbalance ratio (IR). The imbalance ratio, defined as the ratio of majority to minority class instances, serves as a critical indicator of dataset imbalance severity, where higher values signify a more challenging classification task (*Thabtah et al., 2020*). By incorporating datasets with varying levels of class skewness, this study ensures a thorough and robust assessment of the ClusterDEBO method against state-of-the-art oversampling techniques.

All experiments were conducted following the five-fold cross-validation protocol, as defined in the KEEL repository. Each dataset was randomly divided into five equal folds, where four folds were used for training and the remaining fold for testing. This process was repeated five times to ensure that each fold served as the test set once. The final performance metrics were reported as the average of all five runs, minimizing the impact of

**Table 1  The properties of datasets used in the experiments.**

| No | Dataset name | Total samples | Features | Minority class % | Majority class % | Imbalance ratio |
|---|---|---|---|---|---|---|
| 1 | ecoli0137vs26 | 281 | 7 | 2.49 | 97.51 | 39.15 |
| 2 | shuttle0vs4 | 1,829 | 9 | 6.72 | 93.28 | 13.87 |
| 3 | yeastB1vs7 | 459 | 7 | 6.53 | 93.47 | 14.3 |
| 4 | shuttle2vs4 | 129 | 9 | 4.65 | 95.35 | 20.5 |
| 5 | glass016vs2 | 192 | 9 | 8.85 | 91.15 | 10.29 |
| 6 | glass016vs5 | 184 | 9 | 4.89 | 95.11 | 19.44 |
| 7 | pageblocks13vs4 | 472 | 10 | 5.93 | 94.07 | 15.85 |
| 8 | yeast05679vs4 | 528 | 8 | 9.66 | 90.34 | 9.35 |
| 9 | yeast1289vs7 | 947 | 8 | 3.16 | 96.84 | 30.5 |
| 10 | yeast1458vs7 | 693 | 8 | 4.33 | 95.67 | 22.1 |
| 11 | yeast2vs4 | 514 | 8 | 9.92 | 90.08 | 9.08 |
| 12 | Ecoli4 | 336 | 7 | 6.74 | 93.26 | 13.84 |
| 13 | Yeast4 | 1,484 | 8 | 3.43 | 96.57 | 28.41 |
| 14 | Vowel0 | 988 | 13 | 9.01 | 90.99 | 10.1 |
| 15 | Yeast2vs8 | 482 | 8 | 4.15 | 95.85 | 23.1 |
| 16 | Glass4 | 214 | 9 | 6.07 | 93.93 | 15.47 |
| 17 | Glass5 | 214 | 9 | 4.2 | 95.8 | 22.81 |
| 18 | Glass2 | 214 | 9 | 7.94 | 92.06 | 11.59 |
| 19 | Yeast5 | 1,484 | 8 | 2.96 | 97.04 | 32.78 |
| 20 | Yeast6 | 1,484 | 8 | 2.49 | 97.51 | 39.16 |
| 21 | abalone19 | 4,174 | 8 | 0.77 | 99.23 | 128.87 |
| 22 | abalone918 | 731 | 8 | 5.75 | 94.25 | 16.4 |
| 23 | cleveland0vs4 | 177 | 13 | 7.34 | 92.66 | 12.61 |
| 24 | ecoli01vs235 | 244 | 7 | 2.86 | 97.14 | 9.16 |
| 25 | ecoli01vs5 | 240 | 7 | 2.91 | 97.09 | 11 |
| 26 | ecoli0146vs5 | 280 | 7 | 2.5 | 97.5 | 13 |
| 27 | ecoli0147vs2356 | 336 | 7 | 2.08 | 97.92 | 10.58 |
| 28 | ecoli0147vs56 | 332 | 7 | 2.1 | 97.9 | 12.28 |
| 29 | ecoli0234vs5 | 202 | 7 | 3.46 | 96.54 | 9.1 |
| 30 | ecoli0267vs35 | 224 | 7 | 3.12 | 96.88 | 9.18 |
| 31 | ecoli034vs5 | 300 | 7 | 2.33 | 97.67 | 9 |
| 32 | ecoli0346vs5 | 205 | 7 | 3.41 | 96.59 | 9.25 |
| 33 | ecoli0347vs56 | 257 | 7 | 2.72 | 97.28 | 9.28 |
| 34 | ecoli046vs5 | 203 | 7 | 3.44 | 96.56 | 9.15 |
| 35 | ecoli067vs35 | 222 | 7 | 3.15 | 96.85 | 9.09 |
| 36 | ecoli067vs5 | 220 | 7 | 3.18 | 96.82 | 10 |
| 37 | glass0146vs2 | 205 | 9 | 4.39 | 95.61 | 11.05 |
| 38 | glass015vs2 | 172 | 9 | 5.23 | 94.77 | 9.11 |
| 39 | glass04vs5 | 92 | 9 | 9.78 | 90.22 | 9.22 |
| 40 | glass06vs5 | 108 | 9 | 8.33 | 91.67 | 11 |
| 41 | led7digit02456789vs1 | 443 | 7 | 1.58 | 98.42 | 10.97 |

| Table 1 (continued) | | | | | | |
|---|---|---|---|---|---|---|
| No | Dataset name | Total samples | Features | Minority class % | Majority class % | Imbalance ratio |
| 42 | yeast0359vs78 | 506 | 8 | 9.8 | 90.2 | 9.12 |
| 43 | yeast0256vs3789 | 1,004 | 8 | 9.86 | 90.14 | 9.14 |
| 44 | yeast02579vs368 | 1,004 | 8 | 9.86 | 90.14 | 9.14 |

random fluctuations in data splits. This robust validation strategy ensures the reliability and consistency of the evaluation results, providing an unbiased assessment of the proposed method.

In this study, the performance of ClusterDEBO was compared against several widely used oversampling techniques to ensure a comprehensive evaluation. The baseline for comparison was the original dataset without any oversampling. Among the resampling techniques, SMOTE was included as a fundamental benchmark due to its widespread adoption in imbalanced learning. Additionally, Borderline-SMOTE in two variations (Borderline-SMOTE1 and Borderline-SMOTE2) was considered, as these methods focus on generating synthetic samples near the decision boundary to improve classification accuracy. To further strengthen the evaluation, SMOTE-Tomek Links (S-TL) and SMOTE-edited nearest neighbors (S-ENN) were incorporated, combining SMOTE-based oversampling with noise reduction techniques to refine the synthetic sample set. Safe-Level-SMOTE was also included to examine its adaptive approach, which prioritizes generating synthetic samples in safe sample regions. Adaptive Synthetic Sampling (ADASYN) was evaluated as a dynamic resampling technique that adjusts the number of generated samples based on the density of the minority class. Additionally, S-RSB, which employs refined synthetic balancing strategies to enhance minority class representation, was included. Lastly, DEBOHID, an evolutionary algorithm-based oversampling method designed to improve the diversity and spatial distribution of synthetic samples, was tested.

Evaluating classification performance on imbalanced datasets presents unique challenges, as traditional accuracy metrics often fail to provide meaningful insights. In such cases, the area under the curve (AUC) is widely recognized as a robust and reliable measure for assessing a classifier's ability to distinguish between classes.

AUC quantifies the trade-off between the true positive rate (TPR) and false positive rate (FPR) across varying classification thresholds. It is computed using the standard formulation in Eq. (6), where TPR and FPR are defined in Eq. (7).

$$AUC = \frac{1 + TPR - FPR}{2} \tag{6}$$

where:

$$TPR = \frac{TP}{TP + FN}, \quad FPR = \frac{FP}{FP + TN} \tag{7}$$

AUC values range from 0.5 (equivalent to random guessing) to 1.0 (indicating a perfect classifier). Higher AUC values signify better discrimination ability, ensuring that the classifier is effectively distinguishing between the minority and majority classes. This

**Table 2  The mean of the AUC results with the kNN classifier of all methods.**

| Dataset name | Original | | SMOTE | | S-TL | | S-ENN | | Border1 | | Border2 | | Safelevel | | ADASYN | | S-RSB | | DEBOHID | | CLUSTERDEBO | |
|---|---|---|---|---|---|---|---|---|---|---|---|---|---|---|---|---|---|---|---|---|---|---|
| | Mean | Std. dev. | Mean | Std. dev. | Mean | Std. dev. | Mean | Std. dev. | Mean | Std. dev. | Mean | Std. dev. | Mean | Std. dev. | Mean | Std. dev. | Mean | Std. dev. | Mean | Std. dev. | Mean | Std. dev. |
| ecoli0137vs26 | 0.8500 | 0.2236 | 0.8154 | 0.2066 | 0.8118 | 0.2040 | 0.8136 | 0.2040 | 0.8318 | 0.2113 | 0.8391 | 0.2171 | 0.8172 | 0.2064 | 0.8191 | 0.2074 | **0.8581** | 0.2069 | 0.8227 | 0.2042 | 0.8381 | 0.2178 |
| shuttle0vs4 | **0.9960** | 0.0089 | **0.9960** | 0.0089 | **0.9960** | 0.0089 | **0.9960** | 0.0089 | **0.9960** | 0.0089 | **0.9960** | 0.0089 | 0.9951 | 0.0085 | 0.9951 | 0.0085 | **0.9960** | 0.0089 | **0.9960** | 0.0089 | **0.9960** | 0.0089 |
| yeastB1vs7 | 0.5167 | 0.0373 | 0.7156 | 0.0948 | 0.6943 | 0.0698 | 0.7040 | 0.0698 | 0.6417 | 0.1355 | 0.6545 | 0.1146 | 0.6683 | 0.0682 | 0.7121 | 0.0635 | 0.7086 | 0.1150 | **0.7408** | 0.1338 | 0.7402 | 0.0854 |
| shuttle2vs4 | 0.6000 | 0.2236 | 0.9960 | 0.0089 | **1.0000** | 0.0000 | **1.0000** | 0.0000 | **1.0000** | 0.0000 | **1.0000** | 0.0000 | 0.9673 | 0.0374 | 0.9960 | 0.0089 | **1.0000** | 0.0000 | **1.0000** | 0.0000 | **1.0000** | 0.0000 |
| glass016vs2 | 0.5552 | 0.0763 | 0.6686 | 0.0924 | 0.6369 | 0.1310 | 0.6655 | 0.1356 | 0.6893 | 0.1372 | 0.6760 | 0.0540 | 0.6848 | 0.1500 | 0.6493 | 0.0744 | 0.6940 | 0.1342 | 0.6810 | 0.0992 | **0.6960** | 0.0681 |
| glass016vs5 | 0.7386 | 0.1821 | 0.9686 | 0.0156 | 0.9157 | 0.1246 | 0.9186 | 0.1186 | 0.9771 | 0.0078 | 0.8714 | 0.1310 | 0.9457 | 0.0120 | 0.9600 | 0.0256 | 0.9243 | 0.1140 | 0.9243 | 0.1140 | **0.9800** | 0.0137 |
| pageblocks13vs4 | 0.8076 | 0.0998 | 0.9383 | 0.0435 | 0.9250 | 0.0435 | 0.9416 | 0.0735 | 0.9351 | 0.0514 | 0.9351 | 0.0542 | 0.9047 | 0.0306 | 0.9360 | 0.0426 | 0.9106 | 0.0436 | 0.9317 | 0.0446 | 0.9351 | 0.0542 |
| yeast05679vs4 | 0.6667 | 0.0879 | 0.8096 | 0.0607 | 0.8033 | 0.0636 | 0.8138 | 0.0636 | 0.8028 | 0.0684 | 0.8049 | 0.1017 | 0.7974 | 0.0631 | 0.8065 | 0.0554 | 0.7996 | 0.0706 | 0.8313 | 0.0453 | **0.8445** | 0.0714 |
| yeast1289vs7 | 0.4995 | 0.0012 | 0.6322 | 0.1270 | 0.6923 | 0.1270 | 0.6972 | 0.0973 | 0.6107 | 0.1022 | 0.5481 | 0.0323 | 0.6256 | 0.0996 | 0.6617 | 0.1127 | 0.6557 | 0.1047 | 0.6404 | 0.0759 | **0.7294** | 0.0980 |
| yeast1458vs7 | 0.4992 | 0.0017 | 0.6771 | 0.0744 | 0.6824 | 0.0744 | 0.6354 | 0.0786 | 0.5487 | 0.0927 | 0.5032 | 0.0424 | 0.6279 | 0.0682 | 0.6233 | 0.1105 | 0.6430 | 0.0394 | **0.6983** | 0.0658 | 0.6932 | 0.1420 |
| yeast2vs4 | 0.8195 | 0.0452 | 0.8898 | 0.0449 | 0.9055 | 0.0449 | 0.8974 | 0.0149 | 0.8915 | 0.0397 | 0.8725 | 0.0497 | 0.8736 | 0.0230 | 0.8969 | 0.0372 | 0.8709 | 0.0538 | 0.8985 | 0.0306 | **0.9412** | 0.0436 |
| Ecoli4 | 0.8484 | 0.1366 | 0.8965 | 0.0711 | 0.8965 | 0.0734 | 0.8997 | 0.0779 | 0.8905 | 0.0532 | 0.8905 | 0.0577 | 0.8870 | 0.0749 | 0.8870 | 0.0696 | 0.9215 | 0.0795 | 0.9060 | 0.0756 | **0.9393** | 0.0734 |
| Yeast4 | 0.5741 | 0.0508 | 0.7959 | 0.0818 | 0.8238 | 0.0749 | 0.8087 | 0.0497 | 0.7401 | 0.0946 | 0.7554 | 0.0716 | 0.8224 | 0.0494 | 0.8069 | 0.0916 | 0.8069 | 0.0654 | 0.8252 | 0.0825 | **0.8760** | 0.0749 |
| Vowel0 | 0.9772 | 0.0299 | 0.9978 | 0.0023 | 0.9994 | 0.0000 | 0.9994 | 0.0012 | 0.9994 | 0.0012 | 0.9994 | 0.0012 | 0.9783 | 0.0240 | 0.9911 | 0.0042 | 0.9961 | 0.0042 | 0.9994 | 0.0012 | **1.0000** | 0.0000 |
| Yeast2vs8 | 0.7739 | 0.1033 | 0.8295 | 0.1159 | 0.8176 | 0.1159 | 0.8241 | 0.1175 | 0.7502 | 0.1134 | 0.7653 | 0.1094 | 0.7970 | 0.1604 | 0.8372 | 0.1015 | 0.8198 | 0.1258 | 0.7872 | 0.1432 | **0.8966** | 0.1143 |
| Glass4 | 0.7808 | 0.1421 | 0.9001 | 0.0938 | 0.9001 | 0.1078 | 0.9051 | 0.1011 | 0.8917 | 0.1127 | 0.8942 | 0.1144 | 0.8704 | 0.1158 | 0.9052 | 0.1088 | 0.9151 | 0.1000 | 0.9226 | 0.1050 | **0.9338** | 0.1078 |
| Glass5 | 0.6951 | 0.2080 | 0.9537 | 0.0338 | 0.9585 | 0.0159 | 0.9537 | 0.0281 | 0.8780 | 0.2125 | 0.9256 | 0.1268 | 0.9268 | 0.0414 | 0.9463 | 0.0318 | 0.9659 | 0.0291 | 0.8756 | 0.1212 | **0.9902** | 0.0159 |
| Glass2 | 0.4848 | 0.0166 | 0.7948 | 0.0538 | 0.7562 | 0.1000 | 0.7688 | 0.1384 | 0.7001 | 0.0778 | 0.6756 | 0.0734 | 0.7437 | 0.0752 | **0.8029** | 0.1067 | 0.7763 | 0.1079 | 0.7174 | 0.0963 | 0.7906 | 0.1000 |
| Yeast5 | 0.8497 | 0.0627 | 0.9663 | 0.0240 | 0.9649 | 0.0282 | 0.9663 | 0.0232 | 0.9510 | 0.0290 | 0.9500 | 0.0307 | **0.9757** | 0.0055 | 0.9653 | 0.0236 | 0.9656 | 0.0236 | 0.9674 | 0.0261 | 0.9627 | 0.0282 |
| Yeast6 | 0.7387 | 0.1097 | 0.8736 | 0.0738 | 0.8705 | 0.0594 | 0.8705 | 0.0748 | 0.8566 | 0.1053 | 0.8719 | 0.1083 | 0.8749 | 0.0744 | 0.8708 | 0.0765 | 0.8698 | 0.0786 | 0.8367 | 0.0572 | **0.8857** | 0.0594 |
| abalone19 | 0.5000 | 0.0000 | 0.5865 | 0.0702 | 0.5982 | 0.0709 | 0.5817 | 0.0350 | **0.6129** | 0.1051 | 0.5458 | 0.0579 | 0.5939 | 0.0727 | 0.5855 | 0.0707 | 0.5637 | 0.1063 | 0.5734 | 0.0942 | 0.5739 | 0.0709 |
| abalone918 | 0.5681 | 0.0606 | 0.7720 | 0.1107 | 0.7675 | 0.1031 | 0.7703 | 0.1124 | 0.7355 | 0.1403 | 0.6616 | 0.1363 | 0.7663 | 0.1008 | 0.7800 | 0.1155 | 0.8212 | 0.1101 | 0.7815 | 0.1171 | **0.8410** | 0.1031 |
| develand0vs4 | 0.4937 | 0.0086 | 0.5935 | 0.2184 | 0.5621 | 0.1863 | 0.5845 | 0.1891 | 0.6089 | 0.2056 | 0.6247 | 0.2318 | 0.5986 | 0.2418 | 0.5373 | 0.1542 | 0.5499 | 0.1602 | **0.6916** | 0.1912 | 0.5850 | 0.1863 |
| ecoli01vs235 | 0.8300 | 0.0671 | 0.8964 | 0.0475 | 0.8468 | 0.0759 | 0.8941 | 0.1016 | 0.8386 | 0.0707 | 0.8386 | 0.0712 | 0.8450 | 0.0529 | 0.8645 | 0.0811 | 0.8223 | 0.1158 | 0.9036 | 0.0829 | **0.9409** | 0.0759 |
| ecoli01vs5 | 0.9000 | 0.1046 | 0.9045 | 0.0743 | 0.8977 | 0.0697 | 0.9000 | 0.0705 | 0.8932 | 0.1037 | 0.8886 | 0.1014 | 0.8909 | 0.0756 | 0.8614 | 0.1011 | 0.9045 | 0.0790 | 0.9159 | 0.0721 | **0.9182** | 0.0697 |
| ecoli0146vs5 | 0.8981 | 0.1023 | 0.9038 | 0.0978 | 0.9000 | 0.1106 | 0.9058 | 0.1012 | 0.8942 | 0.1008 | 0.8923 | 0.1014 | 0.8962 | 0.1039 | 0.8904 | 0.0977 | 0.9019 | 0.0967 | 0.9173 | 0.1083 | **0.9202** | 0.1106 |
| ecoli0147vs2356 | 0.8467 | 0.0298 | 0.8712 | 0.0235 | 0.8696 | 0.0687 | 0.8760 | 0.0261 | 0.8821 | 0.0603 | 0.9020 | 0.0477 | 0.8329 | 0.0582 | 0.8598 | 0.0206 | 0.8612 | 0.0499 | 0.9122 | 0.0572 | **0.9334** | 0.0687 |
| ecoli0147vs56 | 0.8384 | 0.1509 | 0.8875 | 0.0457 | 0.8711 | 0.0534 | 0.8728 | 0.0356 | 0.9037 | 0.0407 | **0.9069** | 0.0389 | 0.8825 | 0.0388 | 0.8744 | 0.0478 | 0.8776 | 0.0413 | 0.8923 | 0.0429 | 0.9054 | 0.0534 |
| ecoli0234vs5 | 0.8944 | 0.1449 | 0.8975 | 0.1135 | 0.8975 | 0.1191 | 0.9031 | 0.1088 | 0.8890 | 0.1400 | 0.8890 | 0.1400 | 0.8920 | 0.1135 | 0.8561 | 0.1478 | 0.9031 | 0.1141 | 0.9113 | 0.1088 | **0.9167** | 0.1191 |
| ecoli0267vs35 | 0.7875 | 0.0599 | 0.8679 | 0.0595 | 0.8654 | 0.0720 | 0.8654 | 0.0579 | 0.8501 | 0.1106 | 0.8452 | 0.1111 | 0.8079 | 0.1169 | 0.8531 | 0.0583 | 0.8778 | 0.0756 | 0.8728 | 0.0976 | **0.9239** | 0.0720 |
| ecoli034vs5 | 0.8750 | 0.1250 | 0.9028 | 0.1215 | 0.8944 | 0.1145 | 0.9028 | 0.1189 | 0.8944 | 0.1133 | 0.8917 | 0.1073 | 0.8750 | 0.1035 | 0.8583 | 0.1109 | 0.8972 | 0.1165 | 0.9139 | 0.1184 | **0.9174** | 0.1145 |
| ecoli0346vs5 | 0.8750 | 0.0884 | 0.9061 | 0.0647 | 0.8980 | 0.0716 | 0.9088 | 0.0683 | **0.9223** | 0.0711 | 0.9196 | 0.0736 | 0.8845 | 0.0526 | 0.8318 | 0.0873 | 0.9088 | 0.0724 | 0.9142 | 0.0676 | 0.9216 | 0.0716 |
| ecoli0347vs56 | 0.8757 | 0.1337 | 0.8768 | 0.1243 | 0.8768 | 0.1391 | 0.8833 | 0.1241 | 0.9006 | 0.1370 | 0.9006 | 0.1364 | 0.8725 | 0.1175 | 0.8618 | 0.1237 | 0.8769 | 0.1283 | 0.9055 | 0.1271 | **0.9249** | 0.1391 |
| ecoli046vs5 | 0.9000 | 0.1046 | 0.9060 | 0.0972 | 0.9032 | 0.1074 | 0.9061 | 0.0945 | 0.8919 | 0.1052 | 0.8891 | 0.0960 | 0.8868 | 0.0970 | 0.8788 | 0.0970 | 0.9060 | 0.1083 | 0.9142 | 0.1034 | **0.9196** | 0.1074 |
| ecoli067vs35 | 0.8350 | 0.1799 | 0.8975 | 0.1257 | 0.8600 | 0.1369 | 0.8900 | 0.1210 | 0.8300 | 0.1671 | 0.8475 | 0.1664 | 0.8100 | 0.1701 | 0.8800 | 0.1220 | 0.8875 | 0.1259 | 0.8825 | 0.1653 | **0.9010** | 0.1369 |
| ecoli067vs5 | 0.8475 | 0.0548 | 0.8525 | 0.0511 | 0.8575 | 0.0660 | 0.8375 | 0.0641 | **0.9000** | 0.0696 | 0.8975 | 0.0736 | 0.8550 | 0.0338 | 0.8675 | 0.0429 | 0.8675 | 0.0665 | 0.8925 | 0.0699 | 0.8975 | 0.0660 |
| glass0146vs2 | 0.5118 | 0.0635 | **0.7807** | 0.1095 | 0.7367 | 0.1015 | 0.7166 | 0.0967 | 0.6809 | 0.1697 | 0.6883 | 0.0640 | 0.7464 | 0.0855 | 0.7087 | 0.1009 | 0.7246 | 0.0955 | 0.7326 | 0.1059 | 0.7593 | 0.1015 |
| glass015vs2 | 0.5269 | 0.0794 | 0.7427 | 0.1188 | 0.7298 | 0.1004 | 0.7608 | 0.1439 | 0.6462 | 0.1913 | 0.6202 | 0.1343 | 0.7220 | 0.1469 | 0.7427 | 0.1271 | 0.7382 | 0.1258 | 0.7153 | 0.1100 | **0.7815** | 0.1004 |
| glass04vs5 | 0.8500 | 0.1369 | 0.9445 | 0.0558 | 0.9507 | 0.0000 | 0.9570 | 0.0521 | 0.9816 | 0.0278 | 0.9570 | 0.0520 | 0.9154 | 0.0792 | 0.9445 | 0.0558 | 0.9511 | 0.0558 | 0.9820 | 0.0165 | **1.0000** | 0.0000 |

(Continued)

| Dataset name | Original | | SMOTE | | S-TL | | S-ENN | | Border1 | | Border2 | | Safelevel | | ADASYN | | S-RSB | | DEBOHID | | CLUSTERDEBO | |
|---|---|---|---|---|---|---|---|---|---|---|---|---|---|---|---|---|---|---|---|---|---|---|
| | Mean | Std. dev. | Mean | Std. dev. | Mean | Std. dev. | Mean | Std. dev. | Mean | Std. dev. | Mean | Std. dev. | Mean | Std. dev. | Mean | Std. dev. | Mean | Std. dev. | Mean | Std. dev. | Mean | Std. dev. |
| glass06vs5 | 0.7450 | 0.1771 | 0.9750 | 0.0250 | **0.9850** | 0.0137 | **0.9850** | 0.0137 | 0.9350 | 0.1181 | 0.8900 | 0.1399 | 0.9597 | 0.0221 | 0.9697 | 0.0208 | 0.9297 | 0.1150 | 0.9800 | 0.0209 | **0.9850** | 0.0137 |
| led7digit02456789vs1 | 0.5393 | 0.0360 | 0.6386 | 0.1086 | 0.6491 | 0.1057 | **0.8576** | 0.0389 | 0.5881 | 0.1266 | 0.5856 | 0.1271 | 0.5494 | 0.0516 | 0.6361 | 0.1080 | 0.6230 | 0.1165 | 0.6621 | 0.0944 | 0.7087 | 0.1531 |
| yeast0359vs78 | 0.6390 | 0.0660 | 0.7671 | 0.0584 | 0.7406 | 0.0402 | 0.7303 | 0.0393 | 0.7076 | 0.0878 | 0.7007 | 0.0865 | 0.7273 | 0.0671 | 0.7505 | 0.0732 | 0.7283 | 0.0431 | 0.7425 | 0.0801 | **0.7755** | 0.0247 |
| yeast0256vs3789 | 0.7624 | 0.0553 | 0.7781 | 0.0481 | 0.7759 | 0.0480 | 0.7944 | 0.0492 | 0.7643 | 0.0368 | 0.7787 | 0.0302 | 0.7698 | 0.0643 | 0.7798 | 0.0438 | 0.7787 | 0.0684 | 0.7894 | 0.0603 | **0.8281** | 0.0493 |
| yeast02579vs368 | 0.9023 | 0.0260 | 0.9133 | 0.0196 | 0.9016 | 0.0209 | 0.8916 | 0.0310 | 0.8871 | 0.0259 | 0.8971 | 0.0325 | 0.8933 | 0.0273 | 0.8834 | 0.0260 | 0.9044 | 0.0121 | 0.8947 | 0.0224 | **0.9357** | 0.0325 |
| Average | 0.7389 | | 0.8449 | | 0.8383 | | 0.8467 | | 0.8232 | | 0.8156 | | 0.8240 | | 0.8324 | | 0.8387 | | 0.8476 | | 0.8723 | |
| Winner/Total | 1/44 | | 2/44 | | 3/44 | | 5/44 | | 5/44 | | 3/44 | | 1/44 | | 1/44 | | 3/44 | | 5/44 | | 31/44 | |

**Note:**
Bold values denote the highest AUC for each dataset among the compared methods using the kNN classifier. If multiple methods tie for the best AUC on a dataset, all tied values are shown in bold.

characteristic makes AUC particularly well-suited for imbalanced learning tasks, where minimizing bias towards the majority class is critical for real-world applications.

By leveraging AUC as the primary evaluation metric in this study, we provide an objective and consistent comparison of classification performance across different resampling methods, ensuring that improvements are not merely driven by class distribution shifts but by genuine enhancement in predictive capability.

In line with prior research on oversampling, we evaluated the proposed method using classical classifiers kNN, DT, and SVM, which are widely used as standard benchmarks in the field. These models offer a transparent view into the influence of data-level interventions without conflating results with model-specific architectures or hyperparameters. Importantly, our goal was not to maximize classification accuracy *per se*, but to assess whether enhancing the training data *via* ClusterDEBO could yield meaningful improvements even with simpler classifiers. This approach reflects the Garbage In, Garbage Out principle, underscoring that the effectiveness of any classifier, even a deep neural network, depends critically on the quality of the input data. Accordingly, our focus remains on improving data balance and representativeness at the preprocessing stage (*Cinar, 2025*; *Kaya et al., 2021*; *Korkmaz, 2025*).

## RESULTS

In this section, we present a comprehensive analysis of the experimental results obtained using ClusterDEBO and other benchmark oversampling techniques across multiple datasets. The evaluation is performed using three different classifiers: kNN, DT and SVM, ensuring a thorough assessment of the proposed method's performance. First, we provide a comparative analysis of AUC scores across different datasets and classifiers to highlight the effectiveness of ClusterDEBO in handling class imbalance. Next, we conduct a statistical evaluation using the Friedman rank test, demonstrating the statistical significance of the observed performance improvements. In addition to numerical evaluations, visual representations of the datasets before and after applying ClusterDEBO were analyzed to demonstrate the distributional changes introduced by the synthetic data generation process. In this section, we elaborate on key insights and observations from the experimental results, including the advantages of ClusterDEBO, its adaptability across datasets, and its limitations. This analysis provides a holistic understanding of the impact of ClusterDEBO on imbalanced learning tasks.

Table 2 presents the AUC performance results of various oversampling techniques evaluated using the kNN classifier across 44 imbalanced benchmark datasets. The findings reveal that ClusterDEBO achieves the highest average AUC score of 0.8723, outperforming all competing methods. Notably, ClusterDEBO ranks as the best-performing technique in 31 out of 44 datasets, indicating its superior ability to generate well-distributed synthetic samples that enhance classifier performance while preserving class separability.

The baseline model, which utilizes the original dataset without any oversampling, exhibits the lowest mean AUC score of 0.7389, reinforcing the well-documented challenge of class imbalance in machine learning. Traditional oversampling methods such as SMOTE (0.8449), S-TL (0.8383), and S-ENN (0.8467) show moderate improvements but

**Table 3 The mean of the AUC results with the DT classifier of all methods.**

| Dataset name | Original | | SMOTE | | S-TL | | S-ENN | | Border1 | | Border2 | | Safelevel | | ADASYN | | S-RSB | | DEBOHID | | CLUSTERDEBO | |
|---|---|---|---|---|---|---|---|---|---|---|---|---|---|---|---|---|---|---|---|---|---|---|
| | Mean | Std. dev. | Mean | Std. dev. | Mean | Std. dev. | Mean | Std. dev. | Mean | Std. dev. | Mean | Std. dev. | Mean | Std. dev. | Mean | Std. dev. | Mean | Std. dev. | Mean | Std. dev. | Mean | Std. dev. |
| ecoli0137vs26 | 0.8427 | 0.2202 | 0.6818 | 0.2097 | 0.7709 | 0.2104 | 0.7781 | 0.2085 | 0.8427 | 0.2201 | 0.7409 | 0.2501 | 0.7154 | 0.2424 | 0.5800 | 0.1318 | 0.6336 | 0.2197 | 0.7390 | 0.2413 | 0.8409 | 0.2191 |
| shuttle0vs4 | 1.0000 | 0.0000 | 0.9997 | 0.0007 | 1.0000 | 0.0000 | 1.0000 | 0.0000 | 1.0000 | 0.0000 | 1.0000 | 0.0000 | 0.9991 | 0.0013 | 0.9997 | 0.0007 | 0.9997 | 0.0007 | 0.9997 | 0.0007 | 1.0000 | 0.0000 |
| yeastB1vs7 | 0.7100 | 0.0598 | 0.5797 | 0.1179 | 0.6123 | 0.0395 | 0.6844 | 0.1094 | 0.6220 | 0.0738 | 0.6101 | 0.0952 | 0.6459 | 0.1101 | 0.6572 | 0.1111 | 0.6681 | 0.0908 | 0.6383 | 0.0837 | 0.7120 | 0.1382 |
| shuttle2vs4 | 0.9500 | 0.1118 | 0.9918 | 0.0112 | 0.9960 | 0.0089 | 1.0000 | 0.0000 | 0.9500 | 0.1118 | 1.0000 | 0.0000 | 0.9298 | 0.1148 | 0.9960 | 0.0089 | 1.0000 | 0.0000 | 1.0000 | 0.0000 | 1.0000 | 0.0000 |
| glass016vs2 | 0.5548 | 0.0870 | 0.6226 | 0.0625 | 0.6195 | 0.1147 | 0.6421 | 0.0849 | 0.5290 | 0.1101 | 0.6017 | 0.1073 | 0.5819 | 0.1423 | 0.6367 | 0.0834 | 0.7345 | 0.1074 | 0.6071 | 0.1146 | 0.6443 | 0.1161 |
| glass016vs5 | 0.8329 | 0.2299 | 0.8300 | 0.2274 | 0.9329 | 0.1267 | 0.8629 | 0.2588 | 0.8386 | 0.2341 | 0.9386 | 0.1216 | 0.8129 | 0.2366 | 0.8686 | 0.1607 | 0.9214 | 0.1050 | 0.8443 | 0.2229 | 0.8443 | 0.2229 |
| pageblocks13vs4 | 0.9955 | 0.0062 | 0.9475 | 0.0528 | 0.9755 | 0.0372 | 0.9565 | 0.0472 | 0.9978 | 0.0050 | 0.9600 | 0.0501 | 0.9653 | 0.0444 | 0.9354 | 0.0769 | 0.9630 | 0.0443 | 0.9978 | 0.0050 | 0.9824 | 0.0263 |
| yeast05679vs4 | 0.6540 | 0.1137 | 0.8094 | 0.0649 | 0.7585 | 0.0782 | 0.7751 | 0.0911 | 0.7019 | 0.1039 | 0.6821 | 0.0965 | 0.7572 | 0.0493 | 0.7104 | 0.1095 | 0.7441 | 0.0664 | 0.7304 | 0.0918 | 0.7729 | 0.1826 |
| yeast1289vs7 | 0.6353 | 0.1157 | 0.6118 | 0.1115 | 0.6348 | 0.1083 | 0.5870 | 0.1092 | 0.5793 | 0.0459 | 0.6096 | 0.0756 | 0.5856 | 0.0393 | 0.6274 | 0.0998 | 0.6026 | 0.0652 | 0.6389 | 0.0693 | 0.6914 | 0.0948 |
| yeast1458vs7 | 0.5259 | 0.0515 | 0.5025 | 0.0489 | 0.5117 | 0.0660 | 0.4949 | 0.0760 | 0.5448 | 0.0515 | 0.5061 | 0.0401 | 0.5769 | 0.0963 | 0.5783 | 0.1019 | 0.5518 | 0.1001 | 0.6087 | 0.1307 | 0.5795 | 0.0593 |
| yeast2vs4 | 0.8475 | 0.0782 | 0.8428 | 0.0201 | 0.8652 | 0.0550 | 0.9016 | 0.0415 | 0.8353 | 0.0746 | 0.8324 | 0.0693 | 0.8759 | 0.0333 | 0.8369 | 0.0696 | 0.8876 | 0.0447 | 0.8347 | 0.0298 | 0.9197 | 0.0373 |
| Ecoli4 | 0.8624 | 0.1484 | 0.8608 | 0.0806 | 0.8592 | 0.1174 | 0.8278 | 0.1031 | 0.8389 | 0.0971 | 0.8203 | 0.0717 | 0.7997 | 0.1318 | 0.8263 | 0.1374 | 0.8810 | 0.1380 | 0.8389 | 0.0945 | 0.8905 | 0.0981 |
| Yeast4 | 0.6484 | 0.0943 | 0.6965 | 0.0482 | 0.7558 | 0.0767 | 0.6944 | 0.0648 | 0.6754 | 0.0689 | 0.7030 | 0.0710 | 0.7162 | 0.1070 | 0.6712 | 0.0331 | 0.6975 | 0.1122 | 0.7138 | 0.1019 | 0.8881 | 0.0999 |
| Vowel0 | 0.9422 | 0.0513 | 0.9444 | 0.0358 | 0.9727 | 0.0158 | 0.9633 | 0.0430 | 0.9039 | 0.0888 | 0.9533 | 0.0245 | 0.9483 | 0.0441 | 0.9655 | 0.0280 | 0.9589 | 0.0218 | 0.9561 | 0.0449 | 0.9444 | 0.0537 |
| Yeast2vs8 | 0.7696 | 0.1046 | 0.7545 | 0.1322 | 0.7773 | 0.1287 | 0.8066 | 0.1393 | 0.7402 | 0.0924 | 0.7870 | 0.1077 | 0.7796 | 0.1700 | 0.7100 | 0.1638 | 0.7730 | 0.1438 | 0.7762 | 0.1376 | 0.7366 | 0.1566 |
| Glass4 | 0.8567 | 0.1852 | 0.8984 | 0.0954 | 0.8818 | 0.1107 | 0.9392 | 0.0878 | 0.8450 | 0.1570 | 0.9067 | 0.1138 | 0.9051 | 0.1011 | 0.8460 | 0.1670 | 0.8917 | 0.1159 | 0.9350 | 0.1036 | 0.8708 | 0.1102 |
| Glass5 | 0.8427 | 0.2180 | 0.9183 | 0.1042 | 0.9110 | 0.0990 | 0.9207 | 0.1035 | 0.8451 | 0.2230 | 0.7854 | 0.2611 | 0.8037 | 0.2097 | 0.9280 | 0.1015 | 0.8659 | 0.2199 | 0.8976 | 0.2223 | 0.9451 | 0.1092 |
| Glass2 | 0.5376 | 0.1418 | 0.6904 | 0.0508 | 0.6060 | 0.1111 | 0.7132 | 0.1720 | 0.6052 | 0.1301 | 0.6058 | 0.1100 | 0.6429 | 0.1250 | 0.6965 | 0.1092 | 0.8075 | 0.0666 | 0.7390 | 0.0351 | 0.7085 | 0.1558 |
| Yeast5 | 0.8201 | 0.0802 | 0.8521 | 0.0370 | 0.8847 | 0.0272 | 0.9076 | 0.0682 | 0.8337 | 0.0481 | 0.8309 | 0.0546 | 0.9281 | 0.0071 | 0.8885 | 0.0619 | 0.8632 | 0.0268 | 0.9021 | 0.0465 | 0.9532 | 0.0313 |
| Yeast6 | 0.6823 | 0.1109 | 0.7974 | 0.1380 | 0.7936 | 0.1360 | 0.8089 | 0.1203 | 0.7506 | 0.1079 | 0.7649 | 0.1271 | 0.8242 | 0.1415 | 0.7705 | 0.1307 | 0.7953 | 0.0798 | 0.7633 | 0.1136 | 0.8429 | 0.1435 |
| abalone19 | 0.4978 | 0.0020 | 0.5513 | 0.0682 | 0.5308 | 0.0738 | 0.5341 | 0.0746 | 0.4888 | 0.0091 | 0.5087 | 0.0302 | 0.5153 | 0.0408 | 0.5358 | 0.0744 | 0.5283 | 0.0428 | 0.5938 | 0.0431 | 0.5295 | 0.1679 |
| abalone918 | 0.6911 | 0.1521 | 0.7340 | 0.1052 | 0.7340 | 0.1129 | 0.7151 | 0.1001 | 0.6837 | 0.1687 | 0.7274 | 0.1249 | 0.7623 | 0.1253 | 0.7202 | 0.1408 | 0.7411 | 0.1189 | 0.7508 | 0.1189 | 0.7435 | 0.1974 |
| cleveland0vs4 | 0.7888 | 0.1178 | 0.7198 | 0.0674 | 0.7355 | 0.0695 | 0.6833 | 0.0759 | 0.8906 | 0.1506 | 0.7416 | 0.0506 | 0.8282 | 0.0980 | 0.6561 | 0.1256 | 0.7489 | 0.1535 | 0.7139 | 0.0764 | 0.8585 | 0.1185 |
| ecoli01vs235 | 0.8114 | 0.1359 | 0.7709 | 0.1019 | 0.8959 | 0.0704 | 0.8173 | 0.1216 | 0.8136 | 0.1613 | 0.8632 | 0.0950 | 0.7932 | 0.1072 | 0.7936 | 0.0562 | 0.8045 | 0.0641 | 0.8155 | 0.1431 | 0.8363 | 0.1448 |
| ecoli01vs5 | 0.8636 | 0.1512 | 0.8250 | 0.1021 | 0.8159 | 0.0998 | 0.8795 | 0.0915 | 0.8636 | 0.1139 | 0.8614 | 0.1115 | 0.8750 | 0.1134 | 0.7682 | 0.1780 | 0.8727 | 0.0517 | 0.8864 | 0.0927 | 0.8045 | 0.1245 |
| ecoli0146vs5 | 0.7308 | 0.1967 | 0.7981 | 0.1544 | 0.8481 | 0.1645 | 0.8692 | 0.1473 | 0.7904 | 0.1427 | 0.8308 | 0.1021 | 0.7846 | 0.1363 | 0.8731 | 0.1198 | 0.8558 | 0.1207 | 0.8750 | 0.1534 | 0.7952 | 0.2217 |
| ecoli0147vs2356 | 0.8219 | 0.1151 | 0.8174 | 0.0934 | 0.8642 | 0.0578 | 0.8674 | 0.0651 | 0.8071 | 0.0512 | 0.8236 | 0.1525 | 0.8146 | 0.0822 | 0.8293 | 0.1345 | 0.8475 | 0.0668 | 0.8524 | 0.0741 | 0.8863 | 0.0437 |
| ecoli0147vs56 | 0.7886 | 0.1277 | 0.8324 | 0.0654 | 0.7993 | 0.1255 | 0.8122 | 0.0440 | 0.8654 | 0.0858 | 0.8854 | 0.0789 | 0.7923 | 0.0905 | 0.8393 | 0.0921 | 0.8392 | 0.0703 | 0.8837 | 0.0653 | 0.8764 | 0.1150 |
| ecoli0234vs5 | 0.7806 | 0.0624 | 0.8834 | 0.1147 | 0.8892 | 0.1177 | 0.8862 | 0.1098 | 0.8584 | 0.1456 | 0.8195 | 0.1104 | 0.8202 | 0.1222 | 0.8724 | 0.1055 | 0.8562 | 0.1613 | 0.8501 | 0.1262 | 0.8440 | 0.1362 |
| ecoli0267vs35 | 0.7952 | 0.1090 | 0.8577 | 0.0918 | 0.7903 | 0.1283 | 0.8254 | 0.1283 | 0.8078 | 0.1134 | 0.8078 | 0.1165 | 0.7879 | 0.1150 | 0.7754 | 0.1159 | 0.8080 | 0.1132 | 0.8304 | 0.1096 | 0.7990 | 0.1205 |
| ecoli034vs5 | 0.8056 | 0.1562 | 0.8278 | 0.1245 | 0.8667 | 0.1405 | 0.8500 | 0.1308 | 0.7889 | 0.1387 | 0.8111 | 0.1429 | 0.8500 | 0.1143 | 0.8667 | 0.1004 | 0.8972 | 0.1193 | 0.8611 | 0.0997 | 0.8493 | 0.1488 |
| ecoli0346vs5 | 0.8392 | 0.1103 | 0.8676 | 0.0400 | 0.8703 | 0.0448 | 0.8730 | 0.0376 | 0.8446 | 0.1072 | 0.8419 | 0.1356 | 0.8568 | 0.0720 | 0.8345 | 0.0994 | 0.9041 | 0.0433 | 0.8507 | 0.0802 | 0.8014 | 0.0833 |
| ecoli0347vs56 | 0.7692 | 0.0772 | 0.8476 | 0.1544 | 0.8611 | 0.1636 | 0.8675 | 0.1527 | 0.8470 | 0.1480 | 0.8449 | 0.1583 | 0.8325 | 0.1489 | 0.8454 | 0.1115 | 0.9077 | 0.0520 | 0.8834 | 0.0819 | 0.8572 | 0.1564 |
| ecoli046vs5 | 0.8141 | 0.1134 | 0.8119 | 0.1526 | 0.8508 | 0.1151 | 0.8591 | 0.1151 | 0.8336 | 0.1333 | 0.8586 | 0.1290 | 0.8592 | 0.1569 | 0.8924 | 0.0917 | 0.8592 | 0.0863 | 0.8782 | 0.0960 | 0.8816 | 0.0989 |
| ecoli067vs35 | 0.8550 | 0.2181 | 0.8175 | 0.1535 | 0.8125 | 0.1589 | 0.8300 | 0.1589 | 0.7850 | 0.1791 | 0.8275 | 0.1726 | 0.8250 | 0.1635 | 0.8100 | 0.1638 | 0.8225 | 0.1662 | 0.8625 | 0.1556 | 0.9035 | 0.1223 |
| ecoli067vs5 | 0.7700 | 0.2082 | 0.8775 | 0.0681 | 0.8375 | 0.0935 | 0.8900 | 0.0389 | 0.8025 | 0.1857 | 0.8600 | 0.0807 | 0.8650 | 0.0698 | 0.8650 | 0.0907 | 0.8300 | 0.1095 | 0.8275 | 0.1210 | 0.8012 | 0.1352 |
| glass0146vs2 | 0.6120 | 0.1301 | 0.7539 | 0.1298 | 0.7685 | 0.1266 | 0.6757 | 0.0980 | 0.5793 | 0.1159 | 0.5492 | 0.0786 | 0.6751 | 0.1282 | 0.7346 | 0.0326 | 0.7137 | 0.0888 | 0.8137 | 0.1057 | 0.7214 | 0.1663 |
| glass015vs2 | 0.5914 | 0.1430 | 0.6309 | 0.2545 | 0.6796 | 0.2202 | 0.7207 | 0.1659 | 0.6933 | 0.0960 | 0.6419 | 0.1742 | 0.6519 | 0.1706 | 0.7022 | 0.1723 | 0.7304 | 0.1323 | 0.6785 | 0.1960 | 0.6102 | 0.1622 |
| glass04vs5 | 0.9941 | 0.0132 | 0.9401 | 0.0462 | 0.9761 | 0.0247 | 0.9577 | 0.0268 | 0.9941 | 0.0132 | 0.9938 | 0.0140 | 0.9632 | 0.0336 | 0.9574 | 0.0353 | 0.9463 | 0.0237 | 0.9941 | 0.0132 | 0.9824 | 0.0263 |

| Dataset name | Original | | SMOTE | | S-TL | | S-ENN | | Border1 | | Border2 | | Safelevel | | ADASYN | | S-RSB | | DEBOHID | | CLUSTERDEBO | |
|---|---|---|---|---|---|---|---|---|---|---|---|---|---|---|---|---|---|---|---|---|---|---|
| | Mean | Std. dev. | Mean | Std. dev. | Mean | Std. dev. | Mean | Std. dev. | Mean | Std. dev. | Mean | Std. dev. | Mean | Std. dev. | Mean | Std. dev. | Mean | Std. dev. | Mean | Std. dev. | Mean | Std. dev. |
| glass06vs5 | 0.9350 | 0.1055 | 0.9550 | 0.0671 | 0.9647 | 0.0518 | 0.9597 | 0.0517 | 0.9450 | 0.1095 | 0.9897 | 0.0141 | 0.9287 | 0.0354 | 0.9545 | 0.0372 | 0.9800 | 0.0274 | **0.9950** | 0.0112 | 0.9450 | 0.1095 |
| led7digit02456789vs1 | 0.8788 | 0.0740 | 0.8869 | 0.0969 | 0.8808 | 0.0985 | 0.8372 | 0.0810 | 0.8955 | 0.0878 | 0.8931 | 0.0847 | 0.8871 | 0.0443 | 0.8958 | 0.0819 | 0.8689 | 0.0812 | 0.9088 | 0.0725 | **0.9280** | 0.0620 |
| yeast0359vs78 | 0.6804 | 0.0428 | 0.5998 | 0.0760 | 0.6479 | 0.0539 | 0.6365 | 0.0756 | 0.6072 | 0.0399 | 0.6638 | 0.1051 | **0.7268** | 0.0884 | 0.6009 | 0.0225 | 0.6410 | 0.0461 | 0.6534 | 0.0714 | **0.7421** | 0.1183 |
| yeast0256vs3789 | 0.7483 | 0.0280 | 0.7252 | 0.0735 | 0.7402 | 0.0548 | 0.7718 | 0.0262 | 0.7159 | 0.0470 | 0.7056 | 0.0473 | 0.7644 | 0.0563 | 0.7394 | 0.0503 | 0.7193 | 0.0504 | 0.7619 | 0.0731 | **0.7935** | 0.0273 |
| yeast02579vs368 | 0.8715 | 0.0286 | 0.8963 | 0.0421 | **0.9090** | 0.0410 | 0.9168 | 0.0372 | 0.8696 | 0.0190 | 0.8782 | 0.0438 | 0.8910 | 0.0242 | 0.8918 | 0.0460 | 0.8854 | 0.0322 | 0.8724 | 0.0398 | 0.9048 | 0.0715 |
| **Average** | 0.7783 | | 0.7984 | | 0.8111 | | 0.8136 | | 0.7852 | | 0.7924 | | 0.7987 | | 0.7951 | | 0.8147 | | 0.8194 | | **0.8286** | |
| **Winner/Total** | 2/44 | | 3/44 | | 5/44 | | 5/44 | | 4/44 | | 3/44 | | 0/44 | | 1/44 | | 7/44 | | 10/44 | | 16/44 | |

**Note:**
Bold values denote the highest AUC for each dataset among the compared methods using the DT classifier. Ties are bolded.

lack the adaptive mechanisms necessary to optimize sample placement effectively. Borderline-SMOTE variants, which prioritize synthetic sample generation near decision boundaries, yield slightly lower AUC values (0.8232 for Borderline-SMOTE1 and 0.8156 for Borderline-SMOTE2), suggesting that decision-boundary-focused approaches alone may not be sufficient to fully mitigate class overlap and noise accumulation.

Compared to traditional interpolation-based methods, evolutionary oversampling strategies such as DEBOHID (0.8476) demonstrate stronger performance, benefiting from the dynamic optimization of synthetic instances. However, ClusterDEBO surpasses DEBOHID and all other evolutionary approaches by incorporating K-Means clustering, ensuring that synthetic samples are generated in well-defined regions while preventing excessive overlap with the majority class. This targeted sample generation process results in improved classifier robustness and enhanced decision-boundary representation.

Another critical advantage of ClusterDEBO is its stability across datasets, as indicated by its relatively low standard deviation in AUC scores. This consistency can be attributed to its noise reduction and selective sampling mechanisms, which effectively filter out low-quality synthetic instances that may degrade classifier performance. By dynamically adjusting the distribution of generated samples based on intra-cluster relationships, ClusterDEBO prevents synthetic instances from being placed in misleading or ambiguous regions, thereby improving overall classification reliability.

A concrete example illustrating the limitations of SMOTE is observed in the yeast6 dataset. This dataset presents a complex and sparse distribution of the minority class. Here, SMOTE achieves a relatively low AUC of 0.7387 using the kNN classifier, likely due to generating synthetic instances in poorly supported regions that overlap with the majority class. In contrast, ClusterDEBO significantly improves the AUC to 0.8857 by leveraging clustering to localize minority regions and employing DE-based sampling to generate structurally coherent synthetic instances. This example underscores the importance of considering local distribution characteristics and validates the effectiveness of our proposed method in challenging data scenarios.

Table 3 presents the AUC performance of various oversampling methods evaluated using the DT classifier across 44 benchmark datasets. The results demonstrate that ClusterDEBO achieves the highest average AUC score of 0.8286, outperforming all other techniques. Notably, ClusterDEBO is the best-performing method in 16 out of 44 datasets, highlighting its robustness in enhancing minority class classification under the DT model.

The baseline model, which does not apply any oversampling, yields the lowest mean AUC score of 0.7783, further confirming that imbalanced datasets significantly hinder classification performance. While conventional oversampling techniques, such as SMOTE (0.7984) and its derivatives (SMOTE-Tomek Links: 0.8111, SMOTE-ENN: 0.8136), offer moderate improvements, their reliance on linear interpolation limits their ability to generate well-placed synthetic samples, often leading to class overlap and noise accumulation. Similarly, Borderline-SMOTE variants (0.7852 for Borderline-SMOTE1 and 0.7924 for Borderline-SMOTE2) demonstrate inconsistent performance, suggesting that decision boundary-focused approaches alone may not be sufficient for improving classifier robustness.

**Table 4 The mean of the AUC results with the SVM classifier of all methods.**

| Dataset Name | Original | | SMOTE | | S-TL | | S-ENN | | Border1 | | Border2 | | Safelevel | | ADASYN | | S-RSB | | DEBOHID | | CLUSTERDEBO | |
|---|---|---|---|---|---|---|---|---|---|---|---|---|---|---|---|---|---|---|---|---|---|---|
| | Mean | Std. dev. | Mean | Std. dev. | Mean | Std. dev. | Mean | Std. dev. | Mean | Std. dev. | Mean | Std. dev. | Mean | Std. dev. | Mean | Std. dev. | Mean | Std. dev. | Mean | Std. dev. | Mean | Std. dev. |
| ecoli0137vs26 | 0.8500 | 0.2236 | 0.7935 | 0.1959 | 0.7935 | 0.1959 | 0.7971 | 0.2007 | 0.8263 | 0.2118 | 0.8244 | 0.2159 | 0.8327 | 0.2164 | 0.7917 | 0.1949 | 0.8398 | 0.2056 | 0.8172 | 0.1993 | **0.9254** | 0.0717 |
| shuttle0vs4 | **1.0000** | 0.0000 | **1.0000** | 0.0000 | 0.9960 | 0.0000 | **1.0000** | 0.0000 | **1.0000** | 0.0000 | **1.0000** | 0.0000 | 0.9991 | 0.0013 | 0.9994 | 0.0008 | 0.9997 | 0.0007 | **1.0000** | 0.0000 | 0.9920 | 0.0179 |
| yeastB1vs7 | 0.5000 | 0.0000 | 0.7636 | 0.0825 | 0.7543 | 0.0799 | 0.7419 | 0.0479 | 0.6718 | 0.1150 | 0.6707 | 0.1165 | 0.7733 | 0.0670 | 0.7651 | 0.0690 | 0.7605 | 0.0587 | 0.7384 | 0.0482 | **0.7765** | 0.0478 |
| shuttle2vs4 | **1.0000** | 0.0000 | 0.9793 | 0.0361 | **1.0000** | 0.0000 | **1.0000** | 0.0000 | **1.0000** | 0.0000 | **1.0000** | 0.0000 | 0.9470 | 0.0556 | 0.9795 | 0.0292 | **1.0000** | 0.0000 | **1.0000** | 0.0000 | **1.0000** | 0.0000 |
| glass016vs2 | 0.5000 | 0.0000 | 0.5171 | 0.1412 | 0.5407 | 0.1310 | 0.5352 | 0.1023 | 0.6281 | 0.1235 | 0.5979 | 0.0616 | 0.5138 | 0.1022 | 0.5221 | 0.1280 | 0.6293 | 0.0724 | 0.6157 | 0.1116 | 0.5943 | 0.0670 |
| glass016vs5 | 0.4971 | 0.0064 | 0.9486 | 0.0163 | 0.9486 | 0.0217 | 0.9514 | 0.0163 | 0.9800 | 0.0128 | 0.9629 | 0.0278 | 0.9314 | 0.0156 | 0.9514 | 0.0163 | 0.9486 | 0.0128 | 0.9686 | 0.0156 | 0.9200 | 0.0577 |
| pageblocks13vs4 | 0.4728 | 0.1333 | 0.4090 | 0.1235 | 0.4840 | 0.2333 | 0.2948 | 0.1355 | 0.4604 | 0.1287 | 0.4606 | 0.0641 | 0.3069 | 0.1815 | 0.3565 | 0.0655 | 0.3934 | 0.2353 | **0.6004** | 0.1689 | 0.5880 | 0.2053 |
| yeast05679vs4 | 0.5000 | 0.0000 | 0.7865 | 0.0830 | 0.7912 | 0.0612 | 0.7854 | 0.0822 | 0.7943 | 0.0847 | 0.7953 | 0.0862 | 0.7944 | 0.0639 | 0.7829 | 0.0592 | 0.7844 | 0.0803 | 0.7838 | 0.0918 | **0.8362** | 0.0577 |
| yeast1289vs7 | 0.5000 | 0.0000 | 0.7189 | 0.0528 | 0.7263 | 0.0390 | 0.6910 | 0.0390 | 0.6762 | 0.0682 | 0.6757 | 0.0690 | 0.7064 | 0.0532 | 0.7123 | 0.0496 | 0.6946 | 0.0269 | 0.6921 | 0.0343 | **0.7552** | 0.0718 |
| yeast1458vs7 | 0.5000 | 0.0000 | 0.6343 | 0.0519 | 0.6146 | 0.0482 | 0.6357 | 0.0500 | 0.6308 | 0.0662 | 0.6081 | 0.0743 | 0.6328 | 0.0556 | 0.6146 | 0.0433 | 0.6388 | 0.0524 | 0.6333 | 0.0797 | **0.6631** | 0.0658 |
| yeast2vs4 | 0.6691 | 0.1163 | 0.8964 | 0.0331 | 0.8953 | 0.0349 | 0.8907 | 0.0326 | 0.8896 | 0.0293 | 0.8874 | 0.0280 | 0.8885 | 0.0351 | 0.8677 | 0.0253 | 0.8864 | 0.0350 | 0.8805 | 0.0193 | **0.9092** | 0.0273 |
| Ecoli4 | 0.5750 | 0.0685 | 0.9716 | 0.0119 | 0.9402 | 0.0505 | 0.9668 | 0.0103 | 0.9342 | 0.0616 | 0.9295 | 0.0612 | 0.9541 | 0.0206 | 0.9102 | 0.0549 | 0.9620 | 0.0153 | 0.9576 | 0.0526 | **0.9913** | 0.0018 |
| Yeast4 | 0.5000 | 0.0000 | 0.8434 | 0.0217 | 0.8265 | 0.0348 | 0.8338 | 0.0336 | 0.8251 | 0.0300 | 0.8223 | 0.0310 | 0.8286 | 0.0314 | 0.8212 | 0.0355 | 0.8131 | 0.0328 | 0.8104 | 0.0294 | **0.8680** | 0.0332 |
| Vowel0 | 0.8950 | 0.0645 | 0.9699 | 0.0077 | 0.9700 | 0.0095 | 0.9705 | 0.0093 | 0.9200 | 0.0367 | 0.9244 | 0.0560 | 0.9505 | 0.0150 | 0.9616 | 0.0170 | 0.9649 | 0.0080 | 0.9683 | 0.0274 | **0.9858** | 0.0098 |
| Yeast2vs8 | 0.7739 | 0.1033 | 0.7664 | 0.0960 | 0.7653 | 0.0960 | 0.7664 | 0.0960 | 0.7065 | 0.1244 | 0.7141 | 0.1159 | 0.7739 | 0.1033 | 0.7394 | 0.0523 | 0.7631 | 0.0971 | 0.7718 | 0.1023 | **0.8328** | 0.0531 |
| Glass4 | 0.5592 | 0.0993 | 0.9101 | 0.1041 | 0.8977 | 0.1047 | 0.9027 | 0.1076 | **0.9226** | 0.1042 | 0.9176 | 0.1013 | 0.8704 | 0.1091 | 0.9002 | 0.1123 | 0.9002 | 0.1060 | 0.9101 | 0.1052 | 0.8967 | 0.1372 |
| Glass5 | 0.5000 | 0.0000 | 0.9366 | 0.0218 | 0.9366 | 0.0200 | 0.9439 | 0.0185 | **0.9732** | 0.0316 | 0.9561 | 0.0222 | 0.9366 | 0.0200 | 0.9463 | 0.0204 | 0.9415 | 0.0235 | 0.9683 | 0.0329 | 0.9024 | 0.0766 |
| Glass2 | 0.5000 | 0.0000 | 0.6155 | 0.1537 | 0.6777 | 0.0396 | 0.6309 | 0.1197 | 0.6050 | 0.2247 | 0.5880 | 0.2053 | 0.6546 | 0.0812 | 0.6803 | 0.0239 | 0.6212 | 0.1475 | 0.6085 | 0.1486 | **0.6840** | 0.0860 |
| Yeast5 | 0.5000 | 0.0000 | 0.9635 | 0.0066 | 0.9622 | 0.0055 | 0.9642 | 0.0066 | 0.9667 | 0.0033 | 0.9660 | 0.0038 | 0.9608 | 0.0044 | 0.9608 | 0.0068 | 0.9625 | 0.0062 | 0.9635 | 0.0032 | **0.9841** | 0.0049 |
| Yeast6 | 0.5000 | 0.0000 | 0.8730 | 0.0694 | 0.8723 | 0.0694 | 0.8737 | 0.0694 | 0.8791 | 0.0928 | 0.8777 | 0.0922 | 0.8730 | 0.0706 | 0.8628 | 0.0739 | 0.8870 | 0.0574 | 0.8820 | 0.0702 | **0.9192** | 0.0812 |
| abalone19 | 0.5000 | 0.0000 | 0.7453 | 0.0889 | 0.7601 | 0.0889 | 0.7623 | 0.0659 | 0.6914 | 0.1140 | 0.6963 | 0.1528 | 0.7821 | 0.0451 | 0.7786 | 0.0642 | 0.7826 | 0.0869 | **0.7894** | 0.0607 | 0.7618 | 0.0683 |
| abalone918 | 0.5000 | 0.0000 | 0.8581 | 0.0429 | 0.8545 | 0.0419 | 0.8493 | 0.0481 | **0.8833** | 0.0728 | 0.8760 | 0.0769 | 0.8174 | 0.0391 | 0.8530 | 0.0414 | 0.8727 | 0.0360 | 0.8761 | 0.0444 | 0.7832 | 0.0489 |
| cleveland0vs4 | 0.7478 | 0.2172 | 0.9167 | 0.0557 | 0.9167 | 0.0568 | 0.9166 | 0.0655 | 0.7947 | 0.1655 | 0.8426 | 0.1102 | 0.8950 | 0.0652 | 0.9314 | 0.0279 | 0.8735 | 0.0600 | 0.9260 | 0.0645 | **0.9748** | 0.0262 |
| ecoli01vs235 | 0.8359 | 0.1670 | 0.8777 | 0.0981 | 0.8732 | 0.0931 | 0.8845 | 0.1032 | 0.8468 | 0.1525 | 0.8673 | 0.1092 | 0.8555 | 0.0873 | 0.8255 | 0.1379 | 0.8732 | 0.0931 | 0.8918 | 0.1027 | **0.9450** | 0.0704 |
| ecoli01vs5 | 0.8364 | 0.1112 | 0.8591 | 0.0993 | 0.8341 | 0.1239 | 0.8614 | 0.0976 | 0.8864 | 0.1057 | 0.8568 | 0.1257 | 0.8614 | 0.1033 | 0.8068 | 0.1288 | 0.8364 | 0.1267 | 0.8750 | 0.1038 | **0.8886** | 0.1099 |
| ecoli0146vs5 | 0.8635 | 0.1516 | 0.8885 | 0.0956 | 0.8712 | 0.0939 | 0.8519 | 0.0914 | 0.8808 | 0.1595 | 0.8769 | 0.1533 | 0.8769 | 0.0960 | 0.8423 | 0.0614 | 0.8827 | 0.1014 | **0.8981** | 0.1083 | 0.8942 | 0.1270 |
| ecoli0147vs2356 | 0.8267 | 0.0538 | 0.8812 | 0.0710 | 0.8645 | 0.1066 | 0.8661 | 0.0554 | 0.8555 | 0.0470 | 0.8539 | 0.0428 | 0.8796 | 0.0774 | 0.8228 | 0.0509 | 0.8929 | 0.0740 | 0.8658 | 0.0681 | **0.9173** | 0.0579 |
| ecoli0147vs56 | 0.8719 | 0.0863 | 0.8812 | 0.0229 | 0.8730 | 0.0229 | 0.8530 | 0.0704 | 0.8821 | 0.0712 | 0.9021 | 0.0793 | 0.9012 | 0.0376 | 0.8054 | 0.0760 | 0.8779 | 0.0513 | 0.8775 | 0.0608 | **0.9231** | 0.0855 |
| ecoli0234vs5 | 0.8667 | 0.0933 | 0.8891 | 0.1135 | 0.8810 | 0.1133 | 0.8946 | 0.1109 | 0.8806 | 0.1032 | **0.9029** | 0.1134 | 0.8920 | 0.1154 | 0.8204 | 0.1016 | 0.8865 | 0.1108 | 0.9002 | 0.1112 | 0.8730 | 0.1735 |
| ecoli0267vs35 | 0.8526 | 0.1062 | 0.8304 | 0.1198 | 0.8108 | 0.1403 | 0.8604 | 0.1075 | 0.8353 | 0.1020 | 0.8155 | 0.1074 | 0.8507 | 0.1347 | 0.8137 | 0.0904 | **0.8883** | 0.0928 | 0.8379 | 0.1127 | 0.8757 | 0.1403 |
| ecoli034vs5 | 0.8611 | 0.1361 | 0.8611 | 0.1593 | 0.8639 | 0.1606 | 0.8639 | 0.1588 | **0.9333** | 0.0794 | 0.9000 | 0.1160 | 0.8611 | 0.1614 | 0.8111 | 0.1109 | 0.8611 | 0.1593 | 0.8722 | 0.1650 | 0.8917 | 0.1362 |
| ecoli046vs5 | 0.8696 | 0.0838 | 0.8899 | 0.0650 | 0.8926 | 0.0616 | 0.8676 | 0.0324 | 0.8588 | 0.0324 | 0.8588 | 0.0947 | 0.8899 | 0.0549 | 0.7993 | 0.0727 | 0.8872 | 0.0674 | **0.9088** | 0.0724 | 0.8649 | 0.1245 |
| ecoli0347vs56 | 0.8935 | 0.0746 | 0.8947 | 0.0691 | 0.8883 | 0.0719 | 0.8925 | 0.0752 | 0.9007 | 0.0805 | 0.8985 | 0.0846 | 0.9040 | 0.0820 | 0.8651 | 0.0741 | 0.9019 | 0.0843 | 0.9099 | 0.0777 | **0.9215** | 0.1443 |
| ecoli046vs5 | 0.8696 | 0.0887 | 0.8896 | 0.1160 | 0.8843 | 0.1111 | 0.8897 | 0.1143 | 0.8892 | 0.1088 | 0.8809 | 0.1103 | 0.8951 | 0.1158 | 0.8291 | 0.0886 | 0.8788 | 0.1077 | **0.9032** | 0.1170 | 0.9001 | 0.1280 |
| ecoli067vs35 | 0.8525 | 0.2160 | 0.8425 | 0.2032 | 0.8650 | 0.1555 | 0.8525 | 0.2045 | 0.8350 | 0.2094 | 0.8000 | 0.2008 | 0.8400 | 0.2057 | 0.8000 | 0.1970 | 0.8325 | 0.1538 | **0.8700** | 0.1624 | 0.8458 | 0.2633 |
| ecoli067vs5 | 0.8425 | 0.1357 | 0.8525 | 0.0675 | 0.8350 | 0.0757 | 0.8400 | 0.0693 | 0.8825 | 0.0535 | 0.8475 | 0.0768 | 0.8650 | 0.0912 | 0.7800 | 0.0942 | 0.8475 | 0.0681 | 0.8775 | 0.0548 | **0.8912** | 0.1056 |
| glass0146vs2 | 0.5000 | 0.0000 | 0.6067 | 0.0504 | 0.6237 | 0.0759 | 0.6227 | 0.0571 | 0.6277 | 0.0588 | **0.6785** | 0.0687 | 0.6013 | 0.0511 | 0.6174 | 0.0473 | 0.6306 | 0.0881 | 0.6052 | 0.0591 | 0.6775 | 0.0763 |
| glass015vs2 | 0.5000 | 0.0000 | 0.5126 | 0.0685 | 0.5094 | 0.1494 | 0.5320 | 0.0304 | 0.4927 | 0.1287 | 0.5261 | 0.1147 | 0.5191 | 0.0664 | 0.4868 | 0.1023 | 0.5132 | 0.1298 | 0.5000 | 0.0861 | **0.6102** | 0.0596 |
| glass04vs5 | 0.8500 | 0.1369 | 0.9445 | 0.0518 | 0.9449 | 0.0465 | 0.9449 | 0.0465 | **0.9816** | 0.0278 | 0.9570 | 0.0360 | 0.9570 | 0.0415 | 0.9507 | 0.0472 | 0.9691 | 0.0383 | **0.9816** | 0.0278 | 0.9629 | 0.0408 |

(Continued)

| Dataset Name | Original | | SMOTE | | S-TL | | S-ENN | | Border1 | | Border2 | | Safelevel | | ADASYN | | S-RSB | | DEBOHID | | CLUSTERDEBO | |
|---|---|---|---|---|---|---|---|---|---|---|---|---|---|---|---|---|---|---|---|---|---|---|
| | Mean | Std. dev. | Mean | Std. dev. | Mean | Std. dev. | Mean | Std. dev. | Mean | Std. dev. | Mean | Std. dev. | Mean | Std. dev. | Mean | Std. dev. | Mean | Std. dev. | Mean | Std. dev. | Mean | Std. dev. |
| glass06vs5 | 0.6500 | 0.1369 | 0.9387 | 0.0444 | 0.9437 | 0.0439 | 0.9387 | 0.0444 | **0.9689** | 0.0468 | 0.9087 | 0.1159 | 0.9387 | 0.0444 | 0.9387 | 0.0444 | 0.9387 | 0.0444 | 0.9589 | 0.0437 | 0.8834 | 0.0614 |
| led7digit0245678 9vs1 | 0.9056 | 0.0794 | 0.8863 | 0.0579 | 0.8969 | 0.0469 | 0.8760 | 0.0523 | 0.8789 | 0.0444 | 0.8770 | 0.0765 | 0.8883 | 0.0576 | 0.8745 | 0.0729 | 0.8839 | 0.0825 | 0.8748 | 0.0689 | **0.9626** | 0.0282 |
| yeast0359vs78 | 0.6067 | 0.0396 | 0.7484 | 0.0366 | 0.7254 | 0.0395 | 0.7417 | 0.0454 | 0.7482 | 0.0411 | 0.7450 | 0.0474 | 0.7506 | 0.0352 | 0.7025 | 0.0217 | 0.7462 | 0.0383 | 0.7439 | 0.0434 | **0.7913** | 0.0426 |
| yeast0256vs3789 | 0.5486 | 0.0367 | 0.7957 | 0.0421 | 0.7913 | 0.0503 | 0.8007 | 0.0505 | 0.7929 | 0.0563 | 0.7923 | 0.0536 | 0.7940 | 0.0533 | 0.7717 | 0.0397 | 0.7951 | 0.0519 | 0.7907 | 0.0413 | **0.8268** | 0.0716 |
| yeast02579vs368 | 0.8006 | 0.0693 | 0.9007 | 0.0362 | 0.9041 | 0.0320 | 0.9107 | 0.0268 | 0.9107 | 0.0276 | 0.9063 | 0.0268 | 0.9007 | 0.0364 | 0.8638 | 0.0359 | 0.9057 | 0.0302 | 0.9057 | 0.0300 | **0.9279** | 0.0395 |
| Average | 0.6942 | | 0.8293 | | 0.8296 | | 0.8261 | | 0.8279 | | 0.8238 | | 0.8260 | | 0.8072 | | 0.8329 | | **0.8412** | | **0.8595** | |
| Winner/Total | 2/44 | | 1/44 | | 1/44 | | 2/44 | | 8/44 | | 4/44 | | 0/44 | | 0/44 | | 3/44 | | 10/44 | | 26/44 | |

**Note:**
Bold values denote the highest AUC for each dataset among the compared methods using the SVM classifier. Ties are bolded.

Evolutionary oversampling techniques, such as DEBOHID (0.8194) and S-RSB (0.8147), exhibit stronger performance compared to traditional resampling methods, benefiting from more adaptive sample placement strategies. However, ClusterDEBO surpasses all existing methods by incorporating K-Means clustering prior to DE-based sample generation, ensuring that synthetic samples are distributed in regions that enhance class separability while minimizing redundancy. This approach results in more precise decision boundary reinforcement, ultimately improving classification accuracy and model generalization.

Another critical aspect of ClusterDEBO is its stability across different datasets, as reflected by its relatively low standard deviation in AUC scores. Unlike conventional oversampling techniques that generate synthetic instances indiscriminately, ClusterDEBO applies noise reduction and selective sampling strategies, preventing the addition of misleading synthetic samples that could degrade classifier performance. This adaptive sample refinement ensures that only informative instances contribute to model learning, thereby enhancing the overall effectiveness of Decision Tree classifiers.

Table 4 presents the AUC performance results of different oversampling methods across 44 benchmark datasets, demonstrating their impact on classification performance. The ClusterDEBO method achieves the highest overall mean AUC score of 0.8595, significantly outperforming all other approaches. Notably, ClusterDEBO emerges as the best-performing method in 26 out of 44 datasets, reinforcing its effectiveness in handling class imbalance and improving classification robustness.

The original dataset (without any oversampling) yields the lowest average AUC score of 0.6942, emphasizing the necessity of oversampling techniques to enhance classifier performance on imbalanced data. Traditional methods such as SMOTE (0.8293) and SMOTE-ENN (0.8261) offer moderate improvements; however, their reliance on random interpolation often results in overlapping synthetic samples, thereby failing to optimally address decision boundary disparities. Borderline-SMOTE variants (Borderline-SMOTE1: 0.8279 and Borderline-SMOTE2: 0.8238) perform slightly better, but their decision boundary-focused sampling can still introduce noise in certain datasets.

More advanced techniques, such as DEBOHID (0.8412) and S-RSB (0.8329), leverage evolutionary and selective sampling mechanisms to further enhance classification performance. However, ClusterDEBO outperforms these approaches by incorporating a more structured oversampling process through K-Means clustering and DE-based synthetic sample generation. This ensures that synthetic samples are strategically positioned within well-defined clusters, minimizing the risk of noise accumulation while maintaining a clear distinction between minority and majority class distributions.

A key advantage of ClusterDEBO is its consistency across datasets, as demonstrated by its high success rate (26/44 wins) and relatively low standard deviation in performance. Unlike other methods that struggle with dataset-specific characteristics, ClusterDEBO's clustering-driven approach dynamically adapts to underlying data distributions, allowing it to generate well-distributed synthetic samples that align with the natural structure of the dataset.

**Table 5 The overall results of all methods.**

| | | Original | SMOTE | S-TL | S-ENN | Border1 | Border2 | Safelevel | ADASYN | S-RSB | DEBOHID | ClusterDEBO |
|---|---|---|---|---|---|---|---|---|---|---|---|---|
| **kNN AUC** | *Average* | 0.7389 | 0.8449 | 0.8383 | 0.8467 | 0.8232 | 0.8156 | 0.8240 | 0.8324 | 0.8387 | 0.8476 | **0.8723** |
| | *Winner/Total* | 1/44 | 2/44 | 3/44 | 5/44 | 5/44 | 3/44 | 1/44 | 1/44 | 3/44 | 5/44 | **31/44** |
| | *Mean rank* | 9.8409 | 4.8864 | 5.9886 | 4.9432 | 6.8977 | 7.2386 | 7.7159 | 6.8977 | 5.5455 | 4.0795 | **1.9659** |
| | *Final rank* | 11 | 3 | 6 | 4 | 8 | 9 | 10 | 7 | 5 | 2 | **1** |
| | *p-value* | 6.7322e−33 | | | | | | | | | | |
| **DT AUC** | *Average* | 0.7783 | 0.7984 | 0.8111 | 0.8136 | 0.7852 | 0.7924 | 0.7987 | 0.7951 | 0.8147 | 0.9194 | **0.8286** |
| | *Winner/Total* | 2/44 | 3/44 | 5/44 | 5/44 | 4/44 | 3/44 | 0/44 | 1/44 | 7/44 | 10/44 | **16/44** |
| | *Mean rank* | 7.8636 | 6.7045 | 5.1932 | 4.5455 | 7.7955 | 6.7500 | 6.6705 | 6.5455 | 5.2500 | 4.4091 | 4.2727 |
| | *Final rank* | 11 | 8 | 4 | 3 | 10 | 9 | 7 | 6 | 5 | 2 | **1** |
| | *p-value* | 4.4626e−11 | | | | | | | | | | |
| **SVM AUC** | *Average* | 0.6942 | 0.8243 | 0.8296 | 0.8261 | 0.8279 | 0.8238 | 0.8260 | 0.8072 | 0.8329 | 0.8412 | **0.8595** |
| | *Winner/Total* | 2/44 | 1/44 | 1/44 | 2/44 | 8/44 | 4/44 | 0/44 | 0/44 | 3/44 | 10/44 | 26/44 |
| | *Mean rank* | 8.9886 | 5.5909 | 6.4432 | 5.5795 | 5.4432 | 6.1705 | 5.9659 | 8.5795 | 5.4659 | 4.6023 | 3.1705 |
| | *Final rank* | 11 | 6 | 9 | 5 | 3 | 8 | 7 | 10 | 4 | 2 | **1** |
| | *p-value* | 8.7678e−19 | | | | | | | | | | |

**Note:**
Bold highlights the best value(s) within each column. For average AUC and Winner/Total, higher is better; for mean rank, lower is better. Ties are bolded.

The Friedman rank test is a non-parametric statistical test used to detect differences in the performance of multiple algorithms across multiple datasets (*Liu & Xu, 2022*; *Zimmerman & Zumbo, 1993*). It ranks each method for every dataset and then computes the mean rank to determine which method performs better overall. A lower rank indicates a superior method, while a higher rank signifies relatively weaker performance. The *p*-value obtained from the Friedman test evaluates the statistical significance of the observed differences. If the *p*-value is below a predefined threshold (*e.g.*, 0.05), it suggests that the performance differences among the methods are statistically significant.

Table 5 presents the Friedman ranking results for all oversampling methods evaluated across three different classifiers: kNN, DT, and SVM. The table includes the mean AUC scores, the number of datasets where each method achieved the best performance (Winner/Total), the mean rank assigned by the Friedman test, and the final ranking based on these scores.

For the kNN classifier, ClusterDEBO achieves the best overall performance with an average AUC of 0.8723 and a mean rank of 1.9659, significantly outperforming other methods. The *p*-value of $6.73 \times 10^{-33}$ confirms that the observed differences among methods are statistically significant. Traditional oversampling methods such as SMOTE (4.8864 rank), S-ENN (4.9432 rank), and DEBOHID (4.0795 rank) exhibit competitive performance but fall short of ClusterDEBO. The original dataset without oversampling ranks the lowest (rank 9.8409), highlighting the need for effective balancing techniques.

For the Decision Tree (DT) classifier, ClusterDEBO again emerges as the best-performing method, achieving the highest mean AUC (0.8286) and the lowest rank (4.2727), making it the top-ranked method. DEBOHID (4.4091 rank) and
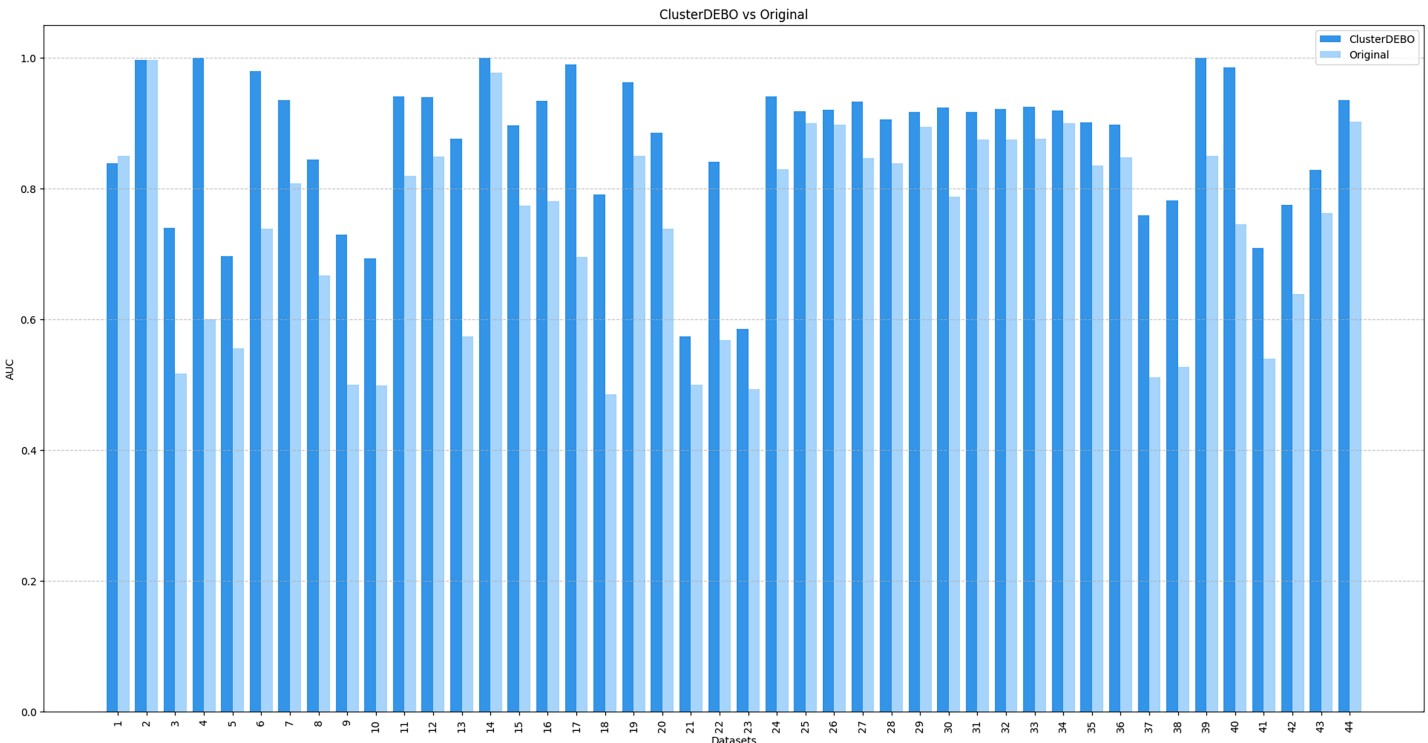

**Figure 7** The comparison of AUC scores for the original datasets and their balanced versions generated by ClusterDEBO, evaluated using the kNN classifier.

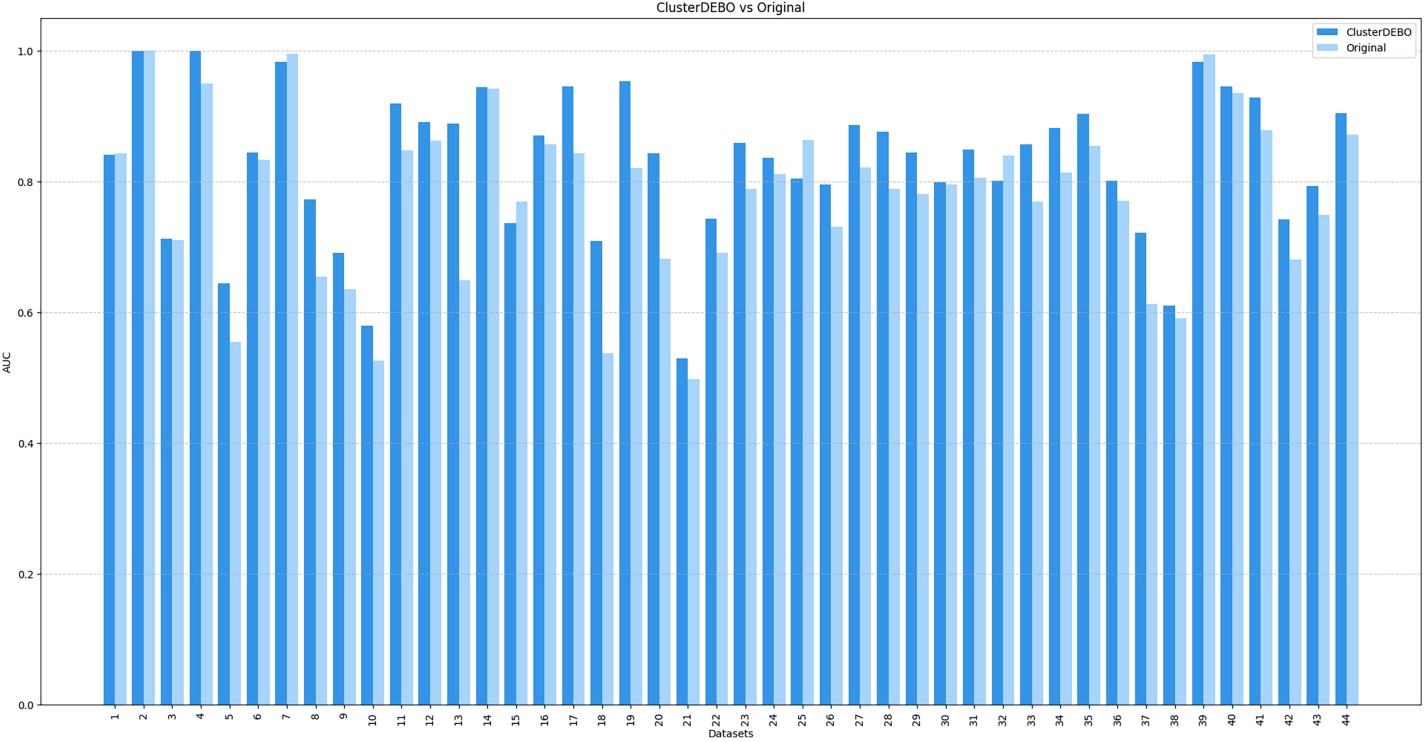

**Figure 8** The comparison of AUC scores for the original datasets and their balanced versions generated by ClusterDEBO, evaluated using the DT classifier.

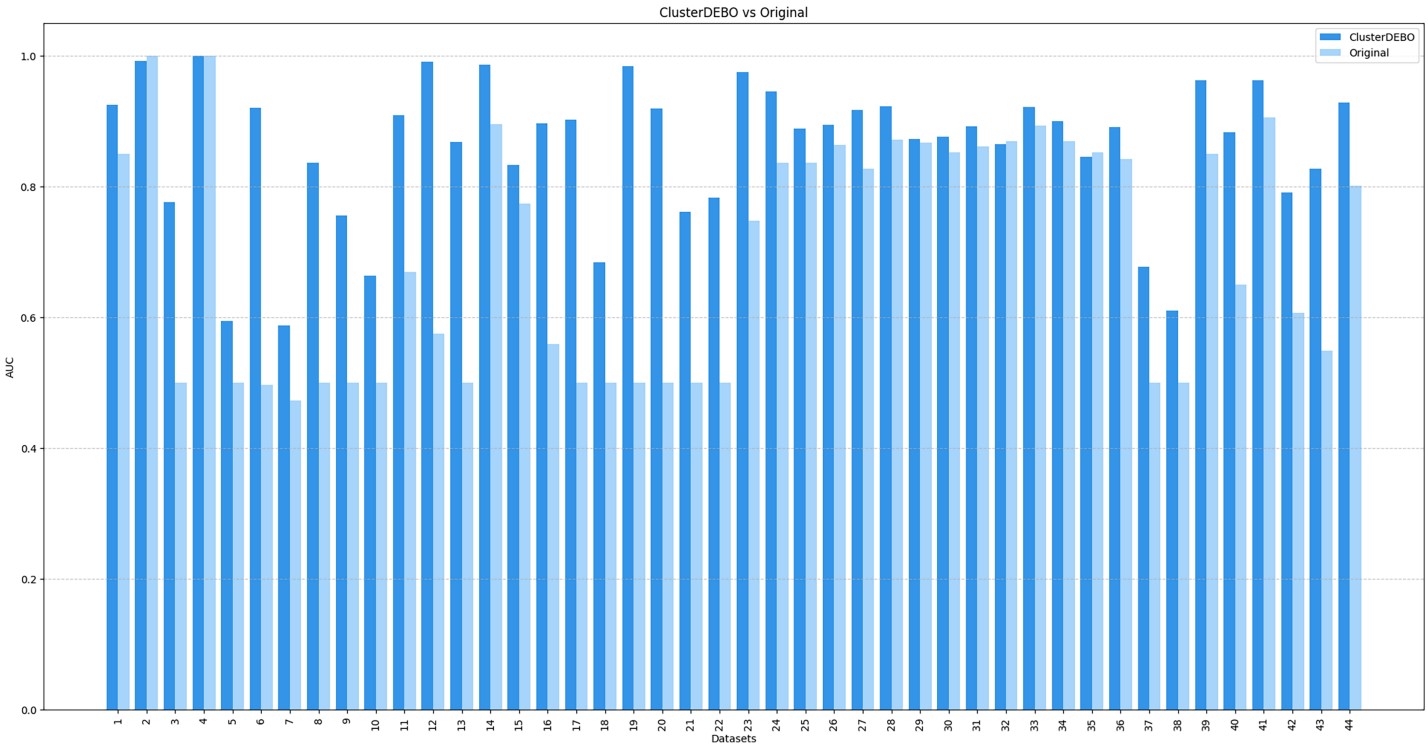

**Figure 9** The comparison of AUC scores for the original datasets and their balanced versions generated by ClusterDEBO, evaluated using the SVM classifier.

S-ENN (4.5455 rank) also show competitive results, suggesting that evolutionary-based techniques provide significant improvements over traditional approaches. The $p$-value $(4.46 \times 10^{-11})$ further confirms that the performance variations across methods are statistically significant.

For the SVM classifier, ClusterDEBO continues to outperform all other methods, ranking first (3.1705 rank) with the highest average AUC (0.8595). DEBOHID (4.6023 rank) and S-RSB (5.4659 rank) also achieve notable performance. The $p$-value $(8.76 \times 10^{-19})$ indicates that the differences among methods are statistically significant. Traditional oversampling techniques such as SMOTE (5.5909 rank) and Borderline-SMOTE (5.4432 rank) provide moderate improvements but do not reach the effectiveness of ClusterDEBO.

The results of the Friedman rank test demonstrate that ClusterDEBO consistently outperforms traditional and evolutionary-based oversampling methods across all three classifiers. It achieves the highest AUC scores and the lowest rank values, confirming its effectiveness in handling imbalanced datasets. The statistically significant $p$-values across all experiments indicate that the observed performance improvements are not due to random variation but are a result of the superior data balancing capability of ClusterDEBO. These findings underscore the potential of integrating clustering and DE strategies for improving classification performance on imbalanced datasets.

Figures 7, 8, and 9 present the comparison of AUC scores for the original datasets and their balanced versions generated by ClusterDEBO, evaluated using the kNN, DT, and SVM classifiers, respectively.

The figure results clearly demonstrate the efficacy of ClusterDEBO in enhancing classification performance across a wide range of datasets.

## DISCUSSION

The results clearly demonstrate that ClusterDEBO is an effective and consistent oversampling method for handling class imbalance across various datasets and classifiers. Its superior performance can be attributed to its structured sample generation strategy that combines K-Means clustering with DE. This integration ensures that synthetic samples are placed in well-defined, informative regions of the feature space, minimizing noise and class overlap.

Unlike traditional methods such as SMOTE and its variants, which rely on random or boundary-based interpolation and often generate samples in ambiguous regions, ClusterDEBO leverages intra-cluster structure to produce cleaner and more meaningful instances. The poor performance of SMOTE in the yeast6 dataset exemplifies this issue, where random interpolation led to overlapping regions. In contrast, ClusterDEBO localized minority regions and generated coherent synthetic samples, dramatically improving AUC.

Furthermore, the method's stability across datasets demonstrated by low standard deviations in performance highlights its adaptability. It effectively filters low-quality samples *via* selective sampling and noise reduction, making it robust against overfitting and performance degradation, a common problem in high-dimensional or noisy imbalanced data.

The results from the Friedman test provide compelling statistical evidence that the observed improvements are not due to chance. Across all three classifiers, ClusterDEBO achieves the lowest ranks and highest AUCs, underscoring its general superiority.

Although recent studies have introduced GAN-based and attention-driven oversampling techniques, many of these methods are not yet integrated into open-source benchmarking libraries that support batch-level evaluations on standardized tabular datasets (*Niu et al., 2023*). For instance, while I-GAN (*Pan et al., 2024*) provides valuable insights and visualizations, its current implementation does not align with the modular and dataset-agnostic framework required for comparative benchmarking across multiple datasets. In contrast, the SMOTE-variants library offers a comprehensive and well-maintained suite of 86 oversampling techniques, enabling transparent, reproducible, and statistically robust evaluations (*Kovács, 2019*). Accordingly, our selection of nine representative methods from this library ensures a meaningful and fair assessment of ClusterDEBO's performance.

By combining structured clustering, adaptive optimization, and careful experimental design, ClusterDEBO presents itself as a practical and statistically validated solution for real-world imbalanced learning scenarios. These findings suggest that further exploration

of hybrid strategies, particularly those that integrate structural learning and evolutionary processes, holds strong potential for advancing the field of imbalanced classification.

## CONCLUSIONS AND FUTURE WORKS

This study introduced ClusterDEBO, a novel oversampling approach that integrates K-Means clustering with DE-based synthetic data generation to address class imbalance in machine learning tasks. By leveraging the clustering step, ClusterDEBO ensures that synthetic samples are generated within meaningful regions of the feature space, while the evolutionary optimization enhances the diversity and representativeness of these samples.

The experimental evaluation, conducted on multiple benchmark datasets across different classifiers (kNN, DT, and SVM), demonstrated that ClusterDEBO significantly improves classification performance compared to traditional and state-of-the-art oversampling techniques. The Friedman rank test results confirmed the statistical superiority of ClusterDEBO, achieving the highest AUC scores and consistently outperforming competing methods. The statistically significant $p$-values across all classifiers further validate its effectiveness, ensuring that the observed improvements are not due to random variation but stem from the robustness of the proposed method.

Furthermore, ClusterDEBO showed superior generalization capabilities, particularly in cases where conventional resampling methods struggled to maintain class boundary integrity. Its ability to adaptively generate synthetic samples that better represent the minority class while preserving decision boundaries offers a promising advancement in imbalanced learning.

Despite its strong performance, several avenues for future research remain open. First, while ClusterDEBO effectively balances imbalanced datasets, its computational complexity may be further optimized by exploring alternative clustering methods or reducing the number of synthetic samples required for improved performance. Hybrid models incorporating deep learning-based feature extraction before applying ClusterDEBO could also be investigated to enhance its adaptability to high-dimensional datasets.

Second, extending ClusterDEBO beyond binary classification to multi-class imbalance scenarios presents an exciting direction for future research. Multi-class settings often involve complex relationships between multiple minority classes, requiring advanced techniques to maintain inter-class boundaries. Developing an adaptive version of ClusterDEBO that dynamically adjusts to varying levels of imbalance across multiple classes would further extend its applicability.

Moreover, although this study fixed the DE parameters and employed the DE/rand/1 (DSt1) strategy, this decision was based on a prior comprehensive study in which 16 DE-based oversampling variants were systematically benchmarked across 44 imbalanced datasets using kNN, DT, and SVM classifiers. That analysis revealed DSt1 as the most effective variant in terms of AUC and G-Mean scores, justifying its direct use in ClusterDEBO without further tuning. Nevertheless, future research could explore dataset-specific or dynamically adaptive tuning of DE parameters, such as mutation factor and crossover rate, through automated search techniques like Bayesian Optimization or

reinforcement learning. Such strategies may yield further improvements by tailoring the synthetic data generation process to the specific distributional characteristics of each dataset, thereby enhancing both performance and flexibility.

Additionally, integrating automated hyperparameter tuning strategies using Bayesian optimization or reinforcement learning could improve the selection of optimal clustering parameters and mutation strategies in DE, ensuring optimal oversampling configurations for diverse datasets. Finally, the practical deployment of ClusterDEBO in real-world applications such as fraud detection, medical diagnosis, and cybersecurity will be an essential next step. Conducting further evaluations on large-scale industrial datasets and integrating ClusterDEBO into end-to-end machine learning pipelines will help assess its real-world utility and scalability. Moreover, belief-based uncertainty modeling approaches such as those grounded in Dempster–Shafer theory may provide complementary mechanisms for identifying reliable regions in the feature space, and their integration with evolutionary oversampling frameworks like ClusterDEBO could be explored to further enhance robustness under high uncertainty.

### Funding
The authors received no funding for this work.

### Competing Interests
The authors declare that they have no competing interests.

### Author Contributions
- Muhammed Abdulhamid Karabiyik conceived and designed the experiments, performed the experiments, analyzed the data, performed the computation work, prepared figures and/or tables, authored or reviewed drafts of the article, and approved the final draft.
- Bahaeddin Turkoglu conceived and designed the experiments, analyzed the data, performed the computation work, prepared figures and/or tables, authored or reviewed drafts of the article, and approved the final draft.
- Tunc Asuroglu conceived and designed the experiments, analyzed the data, authored or reviewed drafts of the article, and approved the final draft.

### Data Availability
The source code for ClusterDEBO is available at Zenodo: bahaeddin turkoglu. (2025). bturkoglu/clusterDEBO: v2 (v2.0). Zenodo. https://doi.org/10.5281/zenodo.15561086.

The datasets used in this study are available from the KEEL repository maintained by the University of Granada: https://sci2s.ugr.es/keel/imbalanced.php#subA.

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
