# Peer review of "A cluster-assisted differential evolution-based hybrid oversampling method for imbalanced datasets"

_PeerJ Computer Science, doi:10.7717/peerj-cs.3177_

## Round 0.1 · original submission · Major Revisions

Reviewer 1 ·

Basic reporting

This paper proposes a hybrid oversampling method that combines K-Means clustering with Differential Evolution (DE) to address class imbalance in machine learning. The following comments are suggestions from my perspective:
1. Although the method of using the Silhouette Score to determine the optimal number of clusters is proposed, I would like to raise a concern: if the minority class contains very few samples, will the clustering process remain stable? Could there be a risk of overfitting in such cases?
2. Differential Evolution involves hyperparameters such as mutation rate and crossover rate. I recommend considering a more systematic investigation into the impact of these parameters on both the quality of synthetic samples and the overall classification performance. Furthermore, I suggest discussing whether dataset-specific adjustments to these parameters might be necessary for optimal results.
3. The proposed method mentions 'low-impact synthetic samples based on their contribution,' but the specific criteria for identifying these samples are not clearly defined. Could you clarify if this is determined based on validation accuracy, or if it is assessed using the distance from the decision boundary of the samples? Additionally, how can you ensure that potentially useful but difficult samples are not mistakenly removed in this process? A more detailed analysis of this aspect would be helpful to understand the robustness of the filtering process.
4. The experimental section currently compares the proposed method only with traditional oversampling techniques. To provide a more comprehensive validation of your approach, I would recommend incorporating some of the latest generative methods for comparison. For example:
[1 ]Conditional Wasserstein GAN-based oversampling of tabular data for imbalanced learning.
[2] An improved generative adversarial network to oversample imbalanced datasets.
[3] Conditional Self-attention generative adversarial network with differential evolution algorithm for imbalanced data classification.
5. The evaluation currently includes only KNN, DT, and SVM. I would suggest considering the inclusion of deeper models, such as lightweight neural networks, for a more thorough assessment. As deep models are becoming increasingly prevalent in modern applications, this could provide further insights into the applicability of ClusterDEBO in real-world scenarios and may influence the conclusions drawn from your study.

Experimental design

-

Validity of the findings

-

·

Basic reporting

This paper is well-structured and generally easy to follow. I believe it meets the standard expectations for such publications. The language used is appropriate and comprehensible, even for non-native English speakers like myself. The figures included are relevant and effectively aid in understanding the content.

However, I have a few minor suggestions for improvement.

Clarify the problem addressed by the new method: It would be beneficial to explicitly highlight the specific issue or phenomenon that the new method aims to address. Providing a concrete example where SMOTE fails — perhaps in scenarios with sparse minority class distributions or complex class boundaries — would help readers understand the motivation behind the new approach.

Enhance Mathematical Formalism in Section 2.2: The notation used in Section 2.2 appears to be based on precise mathematical tools. However, certain expressions, such as max_j, lack clarity due to missing parameters or context. In mathematical formalism, all variables and parameters should be clearly defined to avoid ambiguity. Ensuring that the notation is complete and well-explained will facilitate better understanding and prevent potential confusion.

In summary, while the article is well-written and informative, addressing the above points would enhance its clarity and educational value.

Experimental design

The use of the KEEL dataset repository is standard in studies addressing imbalanced classification problems. However, the outcomes are highly dependent on the specific datasets selected for testing, especially considering that KEEL includes several "problematic" datasets with challenging characteristics such as high overlap between classes or extreme imbalance ratios. To enhance the robustness of the evaluation, it might be beneficial to include synthetic datasets generated . Nonetheless, this is a suggestion rather than a strict requirement.

Regarding performance metrics, the choice of AUC (Area Under the ROC Curve) as the primary evaluation metric warrants reconsideration. While AUC is a popular metric, it may not be the most informative in the context of imbalanced datasets. Metrics such as F1-score and G-mean are often more appropriate. Sensitivity, specificity, and accuracy would facilitate a more comprehensive evaluation and enable comparisons with other published methods that may not have been included in the current study.

The experimental setup and analyses follow standard practices in the field. In this regard, there are no specific comments or concerns.

Validity of the findings

The results presented in the study highlight the advantages of the proposed method, demonstrating its potential effectiveness in addressing the challenges associated with imbalanced classification tasks.

---

## Round 0.2 · Minor Revisions

Reviewer 1 ·

Basic reporting

The revised manuscript presents a well-structured and clearly written study. The proposed ClusterDEBO method effectively combines clustering and differential evolution for oversampling, and the experimental results on 44 benchmark datasets are convincing. The improvements over traditional methods are supported by rigorous statistical validation. Overall, the manuscript is of high quality and is nearly ready for publication.
However, a few minor issues still need to be addressed:
1. Some equations, such as those in the Experimental Setup section, are not numbered. For clarity and easy reference, it is recommended that all equations be appropriately numbered.
2. In recent years, clustering methods based on Dempster-Shafer (D-S) theory have attracted increasing attention for their effectiveness in handling uncertainty and class imbalance, particularly in unsupervised learning and data augmentation tasks. Given that this study involves a clustering-driven oversampling strategy, the authors may consider briefly discussing whether such methods offer potential theoretical or practical insights relevant to the proposed approach.

Experimental design

N/A

Validity of the findings

N/A

Additional comments

N/A

·

Basic reporting

This paper is well-structured and generally easy to follow. I believe it meets the standard expectations for such publications. The language used is appropriate and comprehensible, even for non-native English speakers like myself. The figures included are relevant and effectively aid in understanding the content.
I suggested some modification and extension for improvement. The authors followed my suggestion and they executed the necessary modification of the manuscript.

Experimental design

I suggested using synthetic datasets as well, considering that in such cases we have explicit information about the main characteristics of the dataset (noise level, number of clusters, etc.). I’m glad that the authors would like to involve them in the further work, and I accept that this will be left out of the current paper.

Similarly, I accept the exclusive use of AUC, although I believe that the number of future citations of the paper could be significantly increased if the use of other performance scores allowed the experimental results to be comparable with those of other publications.

Validity of the findings

no comment

---

## Round 0.3 · accepted · Accept

Thanks for undertaking the revision and for your interest in the journal.

Reviewer 1 ·

Basic reporting

-

Experimental design

-

Validity of the findings

-